# ANTI-SYMMETRIC DGN: A STABLE ARCHITECTURE FOR DEEP GRAPH NETWORKS

**Alessio Gravina** *
University of Pisa
alessio.gravina@phd.unipi.it

**Davide Bacciu**
University of Pisa
davide.bacciu@unipi.it

**Claudio Gallicchio**
University of Pisa
claudio.gallicchio@unipi.it

## ABSTRACT

Deep Graph Networks (DGNs) currently dominate the research landscape of learning from graphs, due to their efficiency and ability to implement an adaptive message-passing scheme between the nodes. However, DGNs are typically limited in their ability to propagate and preserve long-term dependencies between nodes, i.e., they suffer from the over-squashing phenomena. This reduces their effectiveness, since predictive problems may require to capture interactions at different, and possibly large, radii in order to be effectively solved. In this work, we present Anti-Symmetric Deep Graph Networks (A-DGNs), a framework for stable and non-dissipative DGN design, conceived through the lens of ordinary differential equations. We give theoretical proof that our method is stable and non-dissipative, leading to two key results: long-range information between nodes is preserved, and no gradient vanishing or explosion occurs in training. We empirically validate the proposed approach on several graph benchmarks, showing that A-DGN leads to improved performance and enables to learn effectively even when dozens of layers are used.

## 1 INTRODUCTION

Representation learning for graphs has become one of the most prominent fields in machine learning. Such popularity derives from the ubiquitousness of graphs. Indeed, graphs are an extremely powerful tool to represent systems of relations and interactions and are extensively employed in many domains (Battaglia et al., 2016; Gilmer et al., 2017; Zitnik et al., 2018; Monti et al., 2019; Derrow-Pinion et al., 2021). For example, they can model social networks, molecular structures, protein-protein interaction networks, recommender systems, and traffic networks.

The primary challenge in this field is how we capture and encode structural information in the learning model. Common methods used in representation learning for graphs usually employ *Deep Graph Networks* (DGNs) (Bacciu et al., 2020; Wu et al., 2021). DGNs are a family of learning models that learn a mapping function that compresses the complex relational information encoded in a graph into an information-rich feature vector that reflects both the topological and the label information in the original graph. As widely popular with neural networks, also DGNs consists of multiple layers. Each of them updates the node representations by aggregating previous node states and their neighbors, following a message passing paradigm. However, in some problems, the exploitation of local interactions between nodes is not enough to learn representative embeddings. In this scenario, it is often the case that the DGN needs to capture information concerning interactions between nodes that are far away in the graph, i.e., by stacking multiple layers. A specific predictive problem typically needs to consider a specific range of node interactions in order to be effectively solved, hence requiring a specific number (possibly large) of DGN layers.

---

*Corresponding author.

Despite the progress made in recent years in the field, many of the proposed methods suffer from the *over-squashing* problem (Alon & Yahav, 2021) when the number of layers increases. Specifically, when increasing the number of layers to cater for longer-range interactions, one observes an excessive amplification or an annihilation of the information being routed to the node by the message passing process to update its fixed length encoding. As such, over-squashing prevents DGNs to learn long-range information.

In this work, we present *Anti-Symmetric Deep Graph Network* (A-DGN), a framework for effective long-term propagation of information in DGN architectures designed through the lens of ordinary differential equations (ODEs). Leveraging the connections between ODEs and deep neural architectures, we provide theoretical conditions for realizing a *stable* and *non-dissipative* ODE system on graphs through the use of anti-symmetric weight matrices. The formulation of the A-DGN layer then results from the forward Euler discretization of the achieved graph ODE. Thanks to the properties enforced on the ODE, our framework preserves the long-term dependencies between nodes as well as prevents from gradient explosion or vanishing. Interestingly, our analysis also paves the way for rethinking the formulation of standard DGNs as discrete versions of non-dissipative and stable ODEs on graphs.

The key contributions of this work can be summarized as follows:

- We introduce A-DGN, a novel design scheme for deep graph networks stemming from an ODE formulation. Stability and non-dissipation are the main properties that characterize our method, allowing the preservation of long-term dependencies in the information flow.
- We theoretically prove that the employed ODE on graphs has stable and non-dissipative behavior. Such result leads to the absence of exploding and vanishing gradient problems during training, typical of unstable and lossy systems.
- We conduct extensive experiments to demonstrate the benefits of our method. A-DGN can outperform classical DGNs over several datasets even when dozens of layers are used.

The rest of this paper is organized as follows. We introduce the A-DGN framework in Section 2 by theoretically proving its properties. In Section 3, we give an overview of the related work in the field of representation learning for graphs and continuous dynamic models. Afterwards, we provide the experimental assessment of our method in Section 4. Finally, Section 5 concludes the paper.

## 2 ANTI-SYMMETRIC DEEP GRAPH NETWORK

Recent advancements in the field of representation learning propose to treat neural network architectures as an ensemble of continuous (rather than discrete) layers, thereby drawing connections between deep neural networks and ordinary differential equations (ODEs) (Haber & Ruthotto, 2017; Chen et al., 2018). This connection can be pushed up to neural processing of graphs as introduced in (Poli et al., 2019), by making a suitable ODE define the computation on a graph structure.

We focus on static graphs, i.e., on structures described by $\mathcal{G} = (\mathcal{V}, \mathcal{E})$, with $\mathcal{V}$ and $\mathcal{E}$ respectively denoting the fixed sets of nodes and edges. For each node $u \in \mathcal{V}$ we consider a state $\mathbf{x}_u(t) \in \mathbb{R}^d$, which provides a representation of the node $u$ at time $t$. We can then define a Cauchy problem on graphs in terms of the following node-wise defined ODE:

$$\frac{\partial \mathbf{x}_u(t)}{\partial t} = f_{\mathcal{G}}(\mathbf{x}_u(t)), \tag{1}$$

for time $t \in [0, T]$, and subject to the initial condition $\mathbf{x}_u(0) = \mathbf{x}_u^0 \in \mathbb{R}^d$. The dynamics of node's representations is described by the function $f_{\mathcal{G}} : \mathbb{R}^d \to \mathbb{R}^d$, while the initial condition $\mathbf{x}_u(0)$ can be interpreted as the initial configuration of the node's information, hence as the input for our computational model. As a consequence, the ODE defined in Equation 1 can be seen as a continuous information processing system over the graph, which starting from the input configuration $\mathbf{x}_u(0)$ computes the final node's representation (i.e., embedding) $\mathbf{x}_u(T)$. Notice that this process shares similarities with standard DGNs, in what it computes nodes' states that can be used as an embedded representation of the graph and then used to feed a readout layer in a downstream task on graphs. The top of Figure 1 visually summarizes this concept, showing how nodes evolve following a specific graph ODE in the time span between 0 and a terminal time $T > 0$.

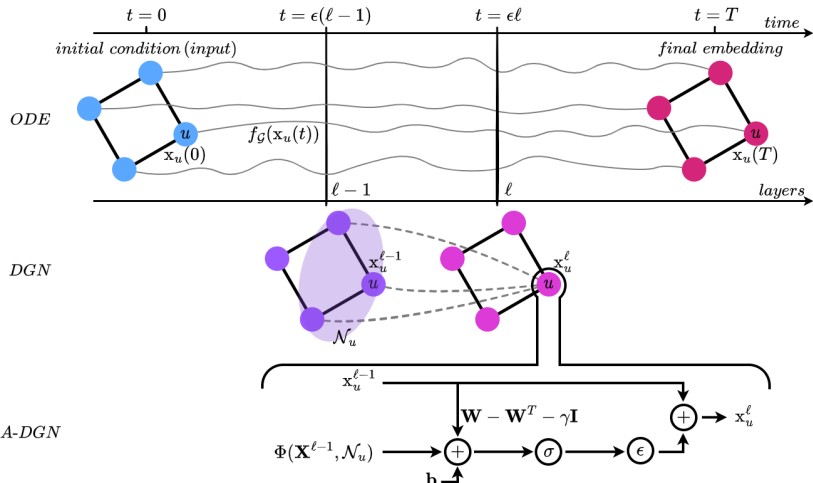

Figure 1: A high level overview of our proposed framework, summarizing the involved concepts of ODE over a graph, its discretization as layers of a DGN, and the resulting node update of Anti-symmetric DGN. At the top, it is illustrated the continuous processing of nodes' states in the time span between 0 and $T > 0$, as a Cauchy problem on graphs. The node-wise ODE $f_{\mathcal{G}}$ determines the evolution of the states $\mathbf{x}_u(t)$, while the initial conditions $\mathbf{x}_u(0)$ play the role of input information. In the middle, the discretized solution of the graph ODE is interpreted as a succession of DGN layers in a neural network architecture. The node state $\mathbf{x}_u^\ell$ computed at layer $\ell$ is updated iteratively by leveraging its neighborhood and self representations at the previous layer $\ell - 1$. The bottom part sketches the computation performed by a layer in an Anti-Symmetric DGN (in Equation 5), resulting from the forward Euler discretization of a stable and non-dissipative ODE on graphs. The more the layers, the more long-range dependencies are included in the final nodes' representations.

Since for most ODEs it is impractical to compute an analytical solution, a common approach relies on finding an approximate one through a numerical discretization procedure (such as the forward Euler method). In this way, the time variable is discretized and the ODE solution is computed by the successive application of an iterated map that operates on the discrete set of points between 0 and $T$, with a step size $\epsilon > 0$. Crucially, as already observed for feed-forward and recurrent neural models (Haber & Ruthotto, 2017; Chang et al., 2019), each step of the ODE discretization process can be equated to one layer of a DGN network. The whole neural architecture contains as many layers as the integration steps in the numerical method (i.e., $L = T/\epsilon$), and each layer $\ell = 1, ..., L$ computes nodes' states $\mathbf{x}_u^\ell$ which approximates $\mathbf{x}_u(\epsilon\,\ell)$. This process is summarized visually in the middle of Figure 1.

Leveraging the concept of graph neural ODEs (Poli et al., 2019), in this paper we perform a further step by reformulating a DGN as a solution to a *stable and non-dissipative* Cauchy problem over a graph. The main goal of our work is therefore achieving preservation of long-range information between nodes, while laying down the conditions that prevent gradient vanishing or explosion. Inspired by the works on stable deep architectures that discretize ODE solutions (Haber & Ruthotto, 2017; Chang et al., 2019), we do so by first deriving conditions under which the graph ODE is constrained to the desired stable and non-dissipative behavior.

Since we are dealing with static graphs, we instantiate Equation 1 for a node $u$ as follows:

$$\frac{\partial \mathbf{x}_u(t)}{\partial t} = \sigma\left(\mathbf{W}_t \mathbf{x}_u(t) + \Phi(\mathbf{X}(t), \mathcal{N}_u) + \mathbf{b}_t\right), \tag{2}$$

where $\sigma$ is a monotonically non-decreasing activation function, $\mathbf{W}_t \in \mathbb{R}^{d \times d}$ and $\mathbf{b}_t \in \mathbb{R}^d$ are, respectively, a weight matrix and a bias vector that contain the trainable parameters of the system. We denote by $\Phi(\mathbf{X}(t), \mathcal{N}_u)$ the aggregation function for the states of the nodes in the neighborhood of $u$. We refer to $\mathbf{X}(t) \in \mathbb{R}^{|\mathcal{V}| \times d}$ as the node feature matrix of the whole graph, with $d$ the number of available features. For simplicity, in the following we keep $\mathbf{W}_t$ and $\mathbf{b}_t$ constant over time, hence dropping the $t$ subscript in the notation.

Well-posedness and stability are essential concepts when designing DGNs as solutions to Cauchy problems, both relying on the continuous dependence of the solution from initial conditions. An ill-posed unstable system, even if potentially yielding a low training error, is likely to lead to a poor generalization error on perturbed data. On the other hand, the solution of a Cauchy problem is stable if the long-term behavior of the system does not depend significantly on the initial conditions (Ascher et al., 1995). In our case, where the ODE defines a message passing diffusion over a graph, our intuition is that a stable encoding system will be robust to perturbations in the input nodes information. Hence, the state representations will change smoothly with the input, resulting in a non-exploding forward propagation and better generalization. This intuition is formalized by the Definition 1 (see Appendix A), whose idea is that a small perturbation of size $\delta$ of the initial state (i.e., the node input features) results in a perturbation on the subsequent states that is at most $\omega$. As known from the stability theory of autonomous systems (Ascher et al., 1995), this condition is met when the maximum real part of the Jacobian's eigenvalues of $f_{\mathcal{G}}$ is smaller or equal than 0, i.e., $\max_{i=1,...,d} Re(\lambda_i(\mathbf{J}(t))) \leq 0, \forall t \geq 0$.

Although stability is a necessary condition for successful learning, it alone is not sufficient to capture long-term dependencies. As it is discussed in Haber & Ruthotto (2017), if $\max_{i=1,...,d} Re(\lambda_i(\mathbf{J}(t))) \ll 0$ the result is a lossy system subject to catastrophic forgetting during propagation. Thus, in the graph domain, this means that only local neighborhood information is preserved by the system, while long-range dependencies among nodes are forgotten. If no long-range information is preserved, then it is likely that the DGN will underperform, since it will not be able to reach the minimum radius of inter-nodes interactions needed to effectively solve the task.

Therefore, we can design an ODE for graphs which is stable and non-dissipative (see Definition 2 in Appendix A) that leads to well-posed learning, when the criterion that guarantees stability is met and the Jacobian's eigenvalues of $f_{\mathcal{G}}$ are nearly zero. Under this condition, the forward propagation produces at most moderate amplification or shrinking of the input, which enables to preserve long-term dependencies in the node states. During training, the backward propagation needed to compute the gradient of the loss $\partial\mathcal{L}/\partial\mathbf{x}_u(t)$ will have the same properties of the forward propagation. As such, no gradient vanish nor explosion is expected to occur. More formally:

**Proposition 1.** *Assuming that $\mathbf{J}(t)$ does not change significantly over time, the forward and backward propagations of the ODE in Equation 2 are stable and non-dissipative if*

$$Re(\lambda_i(\mathbf{J}(t))) = 0, \quad \forall i = 1,...,d. \tag{3}$$

see the proof in Appendix B.

A simple way to impose the condition in Equation 3 is to use an anti-symmetric[1] weight matrix in Equation 2. Under this assumption, we can rewrite Equation 2 as follows:

$$\frac{\partial\mathbf{x}_u(t)}{\partial t} = \sigma\left((\mathbf{W} - \mathbf{W}^T)\mathbf{x}_u(t) + \Phi(\mathbf{X}(t), \mathcal{N}_u) + \mathbf{b}\right) \tag{4}$$

where $(\mathbf{W} - \mathbf{W}^T) \in \mathbb{R}^{d \times d}$ is the anti-symmetric weight matrix. The next Proposition 2 ensures that when the aggregation function $\Phi(\mathbf{X}(t), \mathcal{N}_u)$ is independent of $\mathbf{x}_u(t)$ (see for example Equation 6), the Jacobian of the resulting ODE has imaginary eigenvalues, hence it is stable and non-dissipative according to Proposition 1. As discussed in Appendix D, whenever $\Phi(\mathbf{X}(t), \mathcal{N}_u)$ includes $\mathbf{x}_u(t)$ in its definition (see for example Equation 7), the eigenvalues of the resulting Jacobian are still bounded in a small neighborhood around the imaginary axis.

**Proposition 2.** *Provided that $\Phi(\mathbf{X}(t), \mathcal{N}_u)$ is independent of $\mathbf{x}_u(t)$, the Jacobian matrix of the ODE in Equation 4 has purely imaginary eigenvalues, i.e.*

$$Re(\lambda_i(\mathbf{J}(t))) = 0, \forall i = 1,...,d.$$

*Therefore the ODE in Equation 4 is stable and non-dissipative.*

See proof in Appendix C.

We now proceed to discretize the ODE in Equation 4 by means of the *forward Euler's method*. To preserve stability of the discretized system[2], we add a diffusion term to Equation 4, yielding the

---

[1]A matrix $\mathbf{A} \in \mathbb{R}^{d \times d}$ is anti-symmetric (i.e., skew-symmetric) if $\mathbf{A}^T = -\mathbf{A}$.

[2]The interested reader is referred to (Ascher & Petzold, 1998) for an in-depth analysis on the stability of the forward Euler method.

following node state update equation:

$$\mathbf{x}_u^\ell = \mathbf{x}_u^{\ell-1} + \epsilon\sigma\left((\mathbf{W} - \mathbf{W}^T - \gamma\mathbf{I})\mathbf{x}_u^{\ell-1} + \Phi(\mathbf{X}^{\ell-1}, \mathcal{N}_u) + \mathbf{b}\right) \tag{5}$$

where $\mathbf{I}$ is the identity matrix, $\gamma$ is a hyper-parameter that regulates the strength of the diffusion, and $\epsilon$ is the discretization step. By building on the relationship between the discretization and the DGN layers, we have introduced $\mathbf{x}_u^\ell$ as the state of node $u$ at layer $\ell$, i.e. the discretization of state at time $t = \epsilon\ell$.

Now, both ODE and its Euler discretization are stable and non-dissipative. We refer to the framework defined by Equation 5 as *Anti-symmetric Deep Graph Network* (A-DGN), whose state update process is schematically illustrated in the bottom of Figure 1. Notice that having assumed the parameters of the ODE constant in time, A-DGN can also be interpreted as a recursive DGN with weight sharing between layers.

We recall that $\Phi(\mathbf{X}^{\ell-1}, \mathcal{N}_u)$ can be any function that aggregates nodes (and edges) information. Therefore, the general formulation of $\Phi(\mathbf{X}^{\ell-1}, \mathcal{N}_u)$ in A-DGN allows casting all standard DGNs through in their non-dissipative, stable and well-posed version. As a result, A-DGN can be implemented leveraging the aggregation function that is more adequate for the specific task, while allowing to preserve long-range relationships in the graph. As a demonstration of this, in Section 4 we explore two neighborhood aggregation functions, that are

$$\Phi(\mathbf{X}^{\ell-1}, \mathcal{N}_u) = \sum_{j \in \mathcal{N}_u} \mathbf{V}\mathbf{x}_j^{\ell-1}, \tag{6}$$

(which is also employed in Morris et al. (2019)) and the classical GCN aggregation

$$\Phi(\mathbf{X}^{\ell-1}, \mathcal{N}_u) = \mathbf{V} \sum_{j \in \mathcal{N}_u \cup \{u\}} \frac{1}{\sqrt{\hat{d}_j \hat{d}_u}} \mathbf{x}_j^{\ell-1}, \tag{7}$$

where $\mathbf{V}$ is the weight matrix, $\hat{d}_j$ and $\hat{d}_u$ are, respectively, the degrees of nodes $j$ and $u$.

Finally, although we designed A-DGN with weight sharing in mind (for ease of presentation), a more general version of the framework, with layer-dependent weights $\mathbf{W}^\ell - (\mathbf{W}^\ell)^T$, is possible[3].

## 3 RELATED WORK

**Deep Graph Network**    Nowadays, most of the DGNs typically relies on the concepts introduced by the Message Passing Neural Network (MPNN) (Gilmer et al., 2017), which is a general framework based on the message passing paradigm. The MPNN updates the representation for a node $u$ at layer $\ell$ as

$$\mathbf{x}_u^\ell = \phi_U(\mathbf{x}_u^{\ell-1}, \sum_{j \in \mathcal{N}_u} \phi_M(\mathbf{x}_u^{\ell-1}, \mathbf{x}_v^{\ell-1}, \mathbf{e}_{uv})) \tag{8}$$

where $\phi_U$ and $\phi_M$ are respectively the *update* and *message* functions. Hence, the role of the message function is to compute the message for each node, and then dispatch it among the neighbors. On the other hand, the update function has the role of collecting the incoming messages and update the node state. A typical implementation of the MPNN model is $\mathbf{x}_u^\ell = \mathbf{W}\mathbf{x}_u^{\ell-1} + \sum_{j \in \mathcal{N}_u} \text{MLP}(\mathbf{e}_{uv})\mathbf{x}_v^{\ell-1} + \mathbf{b}$, where $\mathbf{e}_{uv} \in \mathbb{R}^{d_e}$ is the edge feature vector between node $u$ and $v$. Thus, by relaxing the concepts of stability and non-dissipation from our framework, MPNN becomes a specific discretization instance of A-DGN.

Depending on the definition of the update and message functions, it is possible to derive a variety of DGNs that mainly differ on the neighbor aggregation scheme (Kipf & Welling, 2017; Veličković et al., 2018; Hamilton et al., 2017; Xu et al., 2019; Defferrard et al., 2016; Hu et al., 2020). However, all these methods focus on presenting new and effective functions without questioning the stability and non-dissipative behavior of the final network. As a result, most of these DGNs are usually not

---

[3]The dynamical properties discussed in this section are in fact still true even in the case of time varying $\mathbf{W}_t$ in Equation 2, provided that $\max_{i=1,\ldots,d} Re(\lambda_i(\mathbf{J}(t))) \leq 0$ and $\mathbf{J}(t)$ changes sufficiently slow over time (see (Ascher et al., 1995; Haber & Ruthotto, 2017)).

able to capture long-term interactions. Indeed, only few layers can be employed without falling into the over-squashing phenomenon, as it is discussed by Alon & Yahav (2021).

Since the previous methods are all specific cases of MPNN, they are all instances of the discretized and unconstrained version of A-DGN. Moreover, a proper design of $\Phi(\mathbf{X}^{\ell-1}, \mathcal{N}_u)$ in A-DGN allows rethinking the discussed DGNs through the lens of non-dissipative and stable ODEs. Ming Chen et al. (2020), Zhou et al. (2021), and Eliasof et al. (2022) proposed three methods to alleviate over-smoothing, which is a phenomenon where all node features become almost indistinguishable after few embedding updates. Similarly to the forward Euler discretization, the first method employs identity mapping. It also exploits initial residual connections to ensure that the final representation of each node retains at least a fraction of input. The second method proposes a DGN that constrains the Dirichlet energy at each layer and leverages initial residual connections, while the latter tackles over-smoothing by aggregating random paths over the graph nodes. Thus, the novelty of our method is still preserved since A-DGN defines a map between DGNs and stable and non-dissipative graph ODEs to preserve long-range dependencies between nodes.

**Continuous Dynamic Models** Chen et al. (2018) introduce NeuralODE, a new family of neural network models that parametrizes the continuous dynamics of recurrent neural networks using ordinary differential equations. Similarly, Chang et al. (2019) and Gallicchio (2022) draw a connection between ODEs and, respectively, RNNs and Reservoir Computing architectures. Both methods focus on the stability of the solution and the employed numerical method.

Inspired by the NeuralODE approach, Poli et al. (2019) develops a DGN defined as a continuum of layers. In such a work, the authors focus on building the connection between ODEs and DGNs. We extend their work to include stability and non-dissipation, which are fundamental properties to preserve long-term dependencies between nodes and prevent gradient explosion or vanishing during training. Thus, by relaxing these two properties from our framework, the work by Poli et al. (2019) becomes a specific instance of A-DGN. Chamberlain et al. (2021) propose GRAND an architecture to learn graph diffusion as a partial derivative equation (PDE). Differently from GRAND, our framework designs an architecture that is theoretically non-dissipative and free from gradient vanishing or explosion. DGC (Wang et al., 2021) and SGC (Wu et al., 2019) propose linear models that propagate node information as the discretization of the graph heat equation, $\partial \mathbf{X}(t)/\partial t = -\mathbf{L}\mathbf{X}(t)$, without learning. Specifically, DGC mainly focus on exploring the influence of the step size $\epsilon$ in the Euler discretization method. Eliasof et al. (2021) and Rusch et al. (2022) present two methods to preserve the energy of the system, i.e., they mitigate over-smoothing, instead of preserving long-range information between nodes. Differently from our method, which employs a first-order ODE, the former leverages the conservative mapping defined by hyperbolic PDEs, while the latter is defined as second-order ODEs that preserve the Dirichlet energy. In general, this testifies that non-dissipation in graph ODEs is an important property to pursue, not only when preserving long-range dependencies. However, to the best of our knowledge, we are the first to propose a non-dissipative graph ODE to effectively propagate the information on the graph structure.

## 4 EXPERIMENTS

In this section, we discuss the empirical assessment of our method. Specifically, we show the efficacy of preserving long-range information between nodes and mitigating the over-squashing by evaluating our framework on graph property prediction tasks where we predict single source shortest path, node eccentricity, and graph diameter (see Section 4.1). With the same purpose, we report the experiments on the TreeNeighboursMatch problem (Alon & Yahav, 2021) in Appendix H.2. Moreover, we assess the performance of the proposed A-DGN approach on classical graph homophilic (see Section 4.2) and heterophilic (see Appendix H.1) benchmarks. The performance of A-DGN is assessed against DGN variants from the literature. We carried the experiments on a Dell server with 4 Nvidia GPUs A100. We release openly the code implementing our methodology and reproducing our empirical analysis at `https://github.com/gravins/Anti-SymmetricDGN`.

### 4.1 GRAPH PROPERTY PREDICTION

**Setup** For the graph property prediction task, we considered three datasets extracted from the work of Corso et al. (2020). The analysis consists of classical graph theory tasks on undirected

unweighted randomly generated graphs sampled from a wide variety of distributions. Specifically, we considered two node level tasks and one graph level task, which are single source shortest path (SSSP), node eccentricity, and graph diameter. Such tasks require capturing long-term dependencies in order to be solved, thus mitigating the over-squashing phenomenon. Indeed, in the SSSP task, we are computing the shortest paths between a given node $u$ and all other nodes in the graph. Thus, it is fundamental to propagate not only the information of the direct neighborhood of $u$, but also the information of nodes which are extremely far from it. Similarly, for diameter and eccentricity.

We employed the same seed and generator as Corso et al. (2020) to generate the datasets, but we considered graphs with 25 to 35 nodes, instead of 15-25 nodes as in the original work, to increase the task complexity and lengthen long-range dependencies required to solve the task. As in the original work, we used 5120 graphs as training set, 640 as validation set, and 1280 as test set.

We explored the performance of two versions of A-DGN, i.e., weight sharing and layer dependent weights. Moreover, we employed two instances of our method leveraging the two aggregation functions in Equation 6 and 7. We will refer to the former as simple aggregation and to the latter as GCN-based aggregation. We compared our method with GCNII, two neuralODE-based models, i.e., GRAND and DGC, and the four most popular DGNs, i.e., GCN, GAT, GraphSAGE, and GIN.

We designed each model as a combination of three main components. The first is the encoder which maps the node input features into a latent hidden space; the second is the graph convolution (i.e., A-DGN or the DGN baseline); and the third is a readout that maps the output of the convolution into the output space. The encoder and the readout are Multi-Layer Perceptrons that share the same architecture among all models in the experiments.

We performed hyper-parameter tuning via grid search, optimizing the Mean Square Error (MSE). We trained the models using Adam optimizer for a maximum of 1500 epochs and early stopping with patience of 100 epochs on the validation error. For each model configuration, we performed 4 training runs with different weight initialization and report the average of the results.

We report in Appendix F the grid of hyper-parameters exploited for this experiment. We observe that even if we do not directly explore in the hyper-parameter space the terminal time $T$ in which the node evolution produces the best embeddings, that is done indirectly by fixing the values of the step size $\epsilon$ and the maximum number of layers $L$, since $T = L\epsilon$.

**Results** We present the results on the graph property prediction in Table 1. Specifically, we report $log_{10}(\text{MSE})$ as the evaluation metric. We observe that our method, A-DGN, outperforms all the DGNs employed in this experiment. Indeed, by employing GCN-based aggregation, we achieve an error score that is on average 0.80 points better than the selected baselines. Notably, if we leverage the simple aggregation scheme, A-DGN increases the performance gap from the baselines. A-DGN with simple aggregation shows a decisive improvement with respect to baselines. Specifically, it achieves a performance that is 200% to 300% better than the best baseline in each task. Moreover, it is on average $3.6\times$ faster that the baselines (see Table 7 in Appendix G).

We observe that the main challenge when predicting diameter, eccentricity, or SSSP is to leverage not only local information but also global graph information. Such knowledge can only be learned by exploring long-range dependencies. Indeed, the three tasks are extremely correlated. All of them require to compute shortest paths in the graph. Thus, as for standard algorithmic solutions (e.g., Bellman–Ford (Bellman, 1958), Dijkstra's algorithm (Dijkstra, 1959)), more messages between nodes need to be exchanged in order to achieve accurate solutions. This suggests that A-DGN can better capture and exploit such information. Moreover, this indicates also that the simple aggregator is more effective than the GCN-based because the tasks are mainly based on counting distances. Thus, exploiting the information derived from the Laplacian operator is not helpful for solving this kind of algorithmic tasks.

## 4.2 GRAPH BENCHMARKS

**Setup** In the graph benchmark setting we consider five well-known graph datasets for node classification, i.e., PubMed (Namata et al., 2012); coauthor graphs CS and Physics; and the Amazon co-purchasing graphs Computer and Photo from Shchur et al. (2018). Also for this class of experiments, we considered the same baselines and architectural choices as for the graph property

Table 1: Mean test set $log_{10}$(MSE) and std averaged over 4 random weight initializations for each configuration. The lower the better. For each dataset we present the best performing variant of A-DGN, please refer to Apprendix G for the complete results.

|  | Diameter | SSSP | Eccentricity |
|---|---|---|---|
| GCN | 0.7424±0.0466 | 0.9499±9.18·$10^{-5}$ | 0.8468±0.0028 |
| GAT | 0.8221±0.0752 | 0.6951±0.1499 | 0.7909±0.0222 |
| GraphSAGE | 0.8645±0.0401 | 0.2863±0.1843 | 0.7863±0.0207 |
| GIN | 0.6131±0.0990 | -0.5408±0.4193 | 0.9504±0.0007 |
| GCNII | 0.5287±0.0570 | -1.1329±0.0135 | 0.7640±0.0355 |
| DGC | 0.6028±0.0050 | -0.1483±0.0231 | 0.8261±0.0032 |
| GRAND | 0.6715±0.0490 | -0.0942±0.3897 | 0.6602±0.1393 |
| Our | **-0.5455±0.0328** | **-3.4020±0.1372** | **0.3046±0.1181** |
| Our(GCN) | 0.2271±0.0804 | -1.8288±0.0607 | 0.7177±0.0345 |

prediction task. However, in this experiment we study only the version of A-DGN with weight sharing, since it achieve good performances with low training costs.

Within the aim to accurately assess the generalization performance of the models, we randomly split the datasets into multiple train/validation/test sets. Similarly to Shchur et al. (2018), we use 20 labeled nodes per class as the training set, 30 nodes per class as the validation set, and the rest as the test set. We generate 5 random splits per dataset and 5 random weight initialization for each configuration in each split.

We perform hyper-parameter tuning via grid search, optimizing the accuracy score. We train for a maximum of 10000 epochs to minimize the Cross-Entropy loss. We use an early stopping criterion that stops the training if the validation score does not improve for 50 epochs. We report in Appendix F the grid of hyper-parameters explored for this experiment.

**Results**  We present the results on the graph benchmark in Table 2. Specifically, we report the accuracy as the evaluation metric. Even in this scenario, A-DGN outperforms the selected baselines, except in PubMed and Amazon Computers where GCNII is slightly better than our method. In this benchmark, results that the GCN-based aggregation produces higher scores with respect to the simple aggregation. Thus, additional local neighborhood features extracted from the graph Laplacian seem to strengthen the final predictions. It appears also that there is less benefit from including global information with respect to the graph property prediction scenario. As a result, exploiting extremely long-range dependencies do not strongly improve the performance as the number of layers increases.

To demonstrate that our approach performs well with many layers, we show in Figure 2 how the number of layers affects the accuracy score. Our model maintains or improves the performance as the number of layers increases. On the other hand, all the baselines obtain good scores only with one to two layers and most of them exhibit a strong performance degradation as the number of layers increases. Indeed, in the Coauthor CS dataset we obtain that GraphSAGE, GAT, GCN and GIN lose 24.5% to 78.2% of accuracy. We observe that DGC does not degrade its performance since the convolution do not contain parameters.

Although extreme long-range dependencies do not produce the same boost as in the graph property prediction scenario, including more than 5-hop neighborhoods is fundamental to improve state-of-the-art performances. As clear from Figure 2, this is not practical when standard DGNs are employed. On the other hand, A-DGN demonstrates that can capture and exploit such information without any performance drop.

## 5 CONCLUSION

In this paper, we have presented *Anti-Symmetric Deep Graph Network* (A-DGN), a new framework for DGNs achieved from the study of ordinary differential equation (ODE) representing a continuous process of information diffusion on graphs. Unlike previous approaches, by imposing stability and conservative constraints to the ODE through the use of anti-symmetric weight matrices, the proposed framework is able to learn and preserve long-range dependencies between nodes. The A-

Table 2: Mean test set accuracy and std in percent averaged over 5 random train/validation/test splits and 5 random weight initializations for each configuration in each split. The higher the better.

|  | PubMed | Coauthor CS | Coauthor Physics | Amazon Computers | Amazon Photo |
|---|---|---|---|---|---|
| GCN | 76.75±1.29 | 90.34±0.31 | 92.80±0.44 | 81.63±0.93 | 89.14±0.59 |
| GAT | 75.64±1.27 | 81.57±1.02 | 89.25±0.82 | 76.36±0.89 | 85.58±0.91 |
| GraphSAGE | 74.96±1.69 | 89.93±0.79 | 92.47±0.94 | 79.37±1.38 | 88.04±0.85 |
| GIN | 76.24±1.86 | 89.26±0.31 | 91.40±0.70 | 79.64±0.72 | 87.69±1.16 |
| GCNII | **77.39±1.36** | 91.16±0.28 | 92.97±0.60 | **82.72±0.98** | 89.98±0.86 |
| DGC | 66.71±2.55 | 85.84±0.01 | 82.95±1.20 | 66.44±0.63 | 76.13±0.01 |
| GRAND | 76.18±1.56 | 89.20±0.62 | 90.72±0.87 | 81.09±0.70 | 89.05±0.73 |
| Our | 76.57±1.00 | 91.35±0.88 | 92.45±0.53 | 81.83±0.75 | 88.83±1.12 |
| Our(GCN) | 76.82±0.86 | **91.71±0.43** | **93.27±0.62** | 82.35±0.89 | **90.52±0.40** |

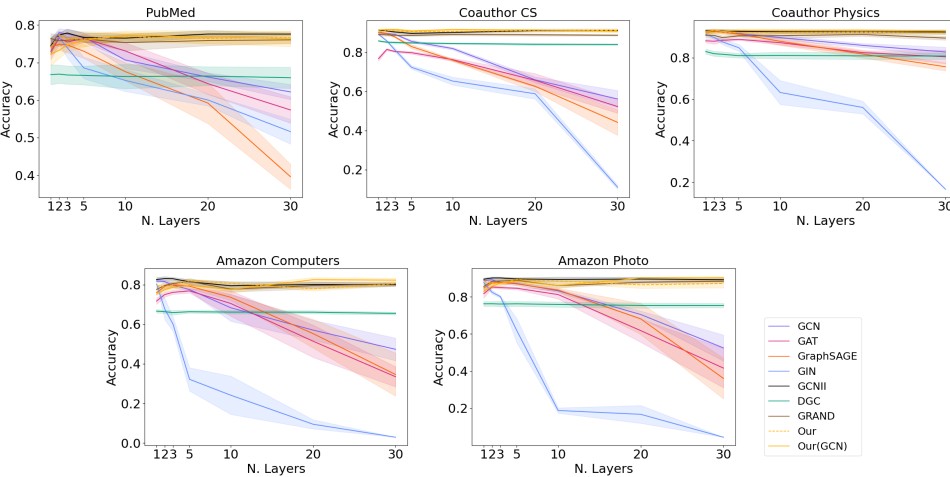

Figure 2: The test accuracy with respect to the number of layers on all the graph benchmark datasets. From the top left to the bottom, we show: PubMed, Coauthor CS, Coauthor Physics, Amazon Computers, and Amazon Photo. The accuracy is averaged over 5 random train/validation/test splits and 5 random weight initialization of the best configuration per split.

DGN layer formulation is the result of the forward Euler discretization of the corresponding Cauchy problem on graphs. We theoretically prove that the differential equation corresponding to A-DGN is stable as well as non-dissipative. Consequently, typical problems of systems with unstable and lossy dynamics, e.g., no gradient explosion or vanishing, do not occur. Thanks to its formulation, A-DGN can be used to reinterpret and extend any classical DGN as a non-dissipative and stable ODEs.

Our experimental analysis, on the one hand, shows that when capturing long-range dependencies is important for the task, our framework largely outperform standard DGNs. On the other hand, it indicates the general competitiveness of A-DGN on several graph benchmarks. Overall, A-DGN shows the ability to effectively explore long-range dependencies and leverage dozens of layers without any noticeable drop in performance. For such reasons, we believe it can be a step towards the mitigation of the over-squashing problem in DGNs.

The results of our experimental analysis and the comparison with state-of-the-art methods suggest that the presented approach can be a starting point to mitigate also over-smoothing. Thus, we plan to accurately explore the consequences of our approach on over-smoothing as a future research direction. Looking ahead to other future developments, we plan to extend the analysis presented in this paper to study DGN architectures resulting from alternative discretization methods of the underlying graph ODE, e.g., using adaptive multi-step schemes (Ascher & Petzold, 1998). Other future lines of investigation include extending the framework to dynamical graph structures, and evaluating its impact in the area of Reservoir Computing (Tanaka et al., 2019).

## ACKNOWLEDGMENTS

This research was partially supported by EMERGE, a project funded by EU Horizon research and innovation programme under GA No 101070918.

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

## A  STABILITY AND DISSIPATIVITY DEFINITIONS

In the following, we define the condition in which the solution of the Cauchy problem in Equation 2 is stable.

**Definition 1.** *A solution $\mathbf{x}_u(t)$ of the ODE in Equation 2, with initial condition $\mathbf{x}_u(0)$, is stable if for any $\omega > 0$, there exists a $\delta > 0$ such that any other solution $\tilde{\mathbf{x}}_u(t)$ of the ODE with initial condition $\tilde{\mathbf{x}}_u(0)$ satisfying $|\mathbf{x}_u(0) - \tilde{\mathbf{x}}_u(0)| \leq \delta$ also satisfies $|\mathbf{x}_u(t) - \tilde{\mathbf{x}}_u(t)| \leq \omega$, for all $t \geq 0$.*

We now provide a definition of a dissipative system based on the one provided in Humphries & Stuart (1994).

**Definition 2.** *Let define $E \subseteq \mathbb{R}^d$ a bounded set that contains any initial condition $\mathbf{x}_u(0)$ for the ODE in Equation 2. The system defined by the ODE in Equation 2 is dissipative if there is a bounded set $B$ where, for any $E$, exists $t^* \geq 0$ such that $\{\mathbf{x}_u(t) \mid \mathbf{x}_u(0) \in E\} \subseteq B$ for $t > t^*$.*

## B  PROOF OF PROPOSITION 1

Let us consider the ODE defined in Equation 2 and analyze the sensitivity of its solution to the initial conditions. Following (Chang et al., 2019), we differentiate both sides of Equation 2 with respect to $\mathbf{x}_u(0)$, obtaining:

$$\frac{d}{dt}\left(\frac{\partial \mathbf{x}_u(t)}{\partial \mathbf{x}_u(0)}\right) = \mathbf{J}(t)\frac{\partial \mathbf{x}_u(t)}{\partial \mathbf{x}_u(0)}. \tag{9}$$

Assuming the Jacobian does not change significantly over time, we can apply results from autonomous differential equations (Glendinning, 1994) and solve Equation 9 analytically as follows:

$$\frac{\partial \mathbf{x}_u(t)}{\partial \mathbf{x}_u(0)} = e^{t\mathbf{J}} = \mathbf{T}e^{t\mathbf{\Lambda}}\mathbf{T}^{-1} = \mathbf{T}\left(\sum_{k=0}^{\infty}\frac{(t\mathbf{\Lambda})^k}{k!}\right)\mathbf{T}^{-1}, \tag{10}$$

where $\mathbf{\Lambda}$ is the diagonal matrix whose non-zero entries contain the eigenvalues of $\mathbf{J}$, and $\mathbf{T}$ has the eigenvectors of $\mathbf{J}$ as columns. The qualitative behavior of $\partial \mathbf{x}_u(t)/\partial \mathbf{x}_u(0)$ is then determined by the real parts of the eigenvalues of $\mathbf{J}$. When $\max_{i=1,\dots,d} Re(\lambda_i(\mathbf{J}(t))) > 0$, a small perturbation of the initial condition (i.e., a perturbation on the input graph) would cause an exponentially exploding difference in the nodes representations, and the system would be unstable. On the contrary, for $\max_{i=1,\dots,d} Re(\lambda_i(\mathbf{J}(t))) < 0$, the term $\partial \mathbf{x}_u(t)/\partial \mathbf{x}_u(0)$ would vanish exponentially fast over time, thereby making the nodes' representation insensitive to differences in the input graph. Accordingly, the system states $\mathbf{x}_u(t)$ would asymptotically approach the same embeddings for all the possible initial conditions $\mathbf{x}_u(0)$, and the system would be dissipative. Notice that the effects of explosion and dissipation are progressively more evident for larger absolute values of $\max_{i=1,\dots,d} Re(\lambda_i(\mathbf{J}(t)))$. If $Re(\lambda_i(\mathbf{J}(t))) = 0$ for $i = 1,\dots,d$ then the magnitude of $\partial \mathbf{x}(t)/\partial \mathbf{x}(0)$ is constant over time, and the input graph information is effectively propagated through the successive transformations into the final nodes' representations. In this last case, the system is hence both stable and non-dissipative.

Let us now consider a loss function $\mathcal{L}$, and observe that its sensitivity to the initial condition (i.e., the input graph) $\partial \mathcal{L}/\partial \mathbf{x}_u(0)$ is proportional to $\partial \mathbf{x}_u(t)/\partial \mathbf{x}_u(0)$. Hence, in light of the previous considerations, if $Re(\lambda_i(\mathbf{J}(t))) = 0$ for $i = 1,\dots,d$, then the magnitude of $\partial \mathcal{L}/\partial \mathbf{x}_u(0)$, which is the longest gradient chain that we can obtain during back-propagation, stays constant over time. The backward propagation is then stable and non-dissipative, and no gradient vanishing or explosion can occur during training.

## C  PROOF OF PROPOSITION 2

Under the assumption that the aggregation function $\Phi(\mathbf{X}(t), \mathcal{N}_u)$ does not include a term that depends on $\mathbf{x}_u(t)$ itself (see Equation 6 for an example), the Jacobian matrix of Equation 4 is given by:

$$\mathbf{J}(t) = \text{diag}\left[\sigma'\left((\mathbf{W} - \mathbf{W}^T)\mathbf{x}_u(t) + \Phi(\mathbf{X}(t), \mathcal{N}_u) + \mathbf{b}\right)\right](\mathbf{W} - \mathbf{W}^T). \tag{11}$$

Following (Chang et al., 2019; 2018), we can see the right-hand side of Equation 11 as the result of a matrix multiplication between an invertible diagonal matrix and an anti-symmetric matrix. Specifically, defining $\mathbf{A} = \text{diag}\left[\sigma'\left((\mathbf{W} - \mathbf{W}^T)\mathbf{x}_u(t) + \Phi(\mathbf{X}(t), \mathcal{N}_u) + \mathbf{b}\right)\right]$ and $\mathbf{B} = \mathbf{W} - \mathbf{W}^T$, we have $\mathbf{J}(t) = \mathbf{AB}$.

Let us now consider an eigenpair of $\mathbf{AB}$, where the eigenvector is denoted by $\mathbf{v}$ and the eigenvalue by $\lambda$. Then:

$$\mathbf{ABv} = \lambda \mathbf{v},$$
$$\mathbf{Bv} = \lambda \mathbf{A}^{-1}\mathbf{v},$$
$$\mathbf{v}^*\mathbf{Bv} = \lambda(\mathbf{v}^*\mathbf{A}^{-1}\mathbf{v}) \tag{12}$$

where $*$ represents the conjugate transpose. On the right-hand side of Equation 12, we can notice that the $(\mathbf{v}^*\mathbf{A}^{-1}\mathbf{v})$ term is a real number. Recalling that $\mathbf{B}^* = \mathbf{B}^T = -\mathbf{B}$ for a real anti-symmetric matrix, we can notice that $(\mathbf{v}^*\mathbf{Bv})^* = \mathbf{v}^*\mathbf{B}^*\mathbf{v} = -\mathbf{v}^*\mathbf{Bv}$. Hence, the $\mathbf{v}^*\mathbf{Bv}$ term on the left-hand side of Equation 12 is an imaginary number. Thereby, $\lambda$ needs to be purely imaginary, and, as a result, all eigenvalues of $\mathbf{J}(t)$ are purely imaginary.

## D    BOUNDED EIGENVALUES OF $\mathbf{J}(t)$

Let us consider here the case in which $\Phi(\mathbf{X}(t), \mathcal{N}_u)$ is defined such that it depends on $\mathbf{x}_u(t)$ (see for example Equation 7). In this case, the Jacobian matrix of Equation 4 can be written (in a more general form than Equation 11), as follows:

$$\mathbf{J}(t) = \mathrm{diag}\left[\sigma'\left((\mathbf{W} - \mathbf{W}^T)\mathbf{x}_u(t) + \Phi(\mathbf{X}(t), \mathcal{N}_u) + \mathbf{b}\right)\right]\left((\mathbf{W} - \mathbf{W}^T) + \mathbf{C}\right), \tag{13}$$

where the term $\mathbf{C}$ represents the derivative of $\Phi(\mathbf{X}(t), \mathcal{N}_u)$ with respect to $\mathbf{x}_u(t)$. For example, for the aggregation function in Equation 7, we have $\mathbf{C} = \mathbf{V}/\hat{d}_u$ (instead, under the assumption of Proposition 2, $\mathbf{C}$ is a zero matrix).

Similarly to Appendix C, we can see the right-hand side of Equation 13 as $\mathbf{J}(t) = \mathbf{A}(\mathbf{B} + \mathbf{C}) = \mathbf{AB} + \mathbf{AC}$. Thereby, we can bound the eigenvalues of $\mathbf{J}(t)$ around those of $\mathbf{AB}$ by applying the results of the Bauer-Fike's theorem (Bauer & Fike, 1960). Recalling that the eigenvalues of $\mathbf{AB}$ are all imaginary (as proved in Appendix C), we can conclude that the eigenvalues of $\mathbf{J}(t)$ are contained in a neighborhood of the imaginary axis with radius $r = \|\mathbf{AC}\| \leq \|\mathbf{C}\|$. For example, using the definition of the aggregation function $\Phi(\mathbf{X}(t), \mathcal{N}_u)$ given in Equation 7, we have $r \leq \|\mathbf{V}\|/\hat{d}_u$.

Although this result does not guarantee that the eigenvalues of the Jacobian are imaginary, in practice it crucially limits their position around the imaginary axis, limiting the dynamics of the system on the graph to show at most moderate amplification or loss of signals over the structure.

## E    DATASETS DESCRIPTION AND STATISTICS

In the graph property prediction (GPP) experiments, we employed the same generation procedure as in Corso et al. (2020). Graphs are randomly sampled from several graph distributions, such as Erdős–Rényi, Barabasi-Albert, and grid. Each node have random identifiers as input features. Target values represent single source shortest path, node eccentricity, and graph diameter.

PubMed is a citation network where each node represents a paper and each edge indicates that one paper cites another one. Each publication in the dataset is described by a 0/1-valued word vector indicating the absence/presence of the corresponding word from the dictionary. The class labels represent the papers categories.

Amazon Computers and Amazon Photo are portions of the Amazon co-purchase graph, where nodes represent goods and edges indicate that two goods are frequently bought together. Node features are bag-of-words encoded product reviews, and class labels are given by the product category.

Coauthor CS and Coauthor Physics are co-authorship graphs extracted from the Microsoft Academic Graph[4] where nodes are authors, that are connected by an edge if they co-authored a paper. Node features represent paper keywords for each author's papers, and class labels indicate most active fields of study for each author.

Table 3 contains the statistics of the employed datasets, sorted by graph density. The density of a graph is computed as the ratio between the number of edges and the number of possible edges, i.e., $d = \frac{|\mathcal{E}|}{|\mathcal{V}|(|\mathcal{V}|-1)}$.

---

[4]https://www.kdd.org/kdd-cup/view/kdd-cup-2016/Data

Table 3: Datasets statistics ordered by graph density.

|  | Nodes | Edges | Features | Classes | Density |
|---|---|---|---|---|---|
| GPP | 25 - 35 | 22 - 553 | 2 | — | 0.0275 - 0.5 |
| Amazon Computers | 13,752 | 491,722 | 767 | 10 | $2.6e^{-3}$ |
| Amazon Photo | 7,650 | 238,162 | 745 | 8 | $4.1e^{-3}$ |
| PubMed | 19,717 | 88,648 | 500 | 3 | $2.3e^{-4}$ |
| Coauthor Physics | 34,493 | 495,924 | 8,415 | 5 | $4.2e^{-4}$ |
| Coauthor CS | 18,333 | 163,788 | 6,805 | 15 | $4.9e^{-4}$ |

# F    EXPLORED HYPER-PARAMETER SPACE

In Table 4 and Table 5 we report the grids of hyper-parameters employed in our experiments by each method. We recall that the hyper-parameters $\epsilon$ and $\gamma$ refer only to our method.

Table 4: The grid of hyper-parameters employed during model selection for the graph property prediction task.

|  | Configs |
|---|---|
| optimizer | Adam |
| learning rate | 0.003 |
| weight decay | $10^{-6}$ |
| n. layers | 1, 5, 10, 20 |
| embedding dim | 10, 20, 30 |
| $\sigma$ | tanh |
| $\epsilon$ | $1, 10^{-1}, 10^{-2}, 10^{-3}$ |
| $\gamma$ | $1, 10^{-1}, 10^{-2}, 10^{-3}$ |

Table 5: The grid of hyper-parameters employed during model selection in the graph benchmark.

|  | Configs |
|---|---|
| optimizer | AdamW |
| learning rate | $10^{-2}, 10^{-3}, 10^{-4}$ |
| weight decay | 0.1 |
| n. layers | 1, 2, 3, 5 ,10, 20, 30 |
| embedding dim | 32, 64, 128 |
| $\sigma$ | tanh |
| $\epsilon$ | $1, 10^{-1}, 10^{-2}, 10^{-3}, 10^{-4}$ |
| $\gamma$ | $1, 10^{-1}, 10^{-2}, 10^{-3}, 10^{-4}$ |

# G    COMPLETE RESULTS

In Table 6 we report the complete results for the graph property prediction benchmark, including all the versions and cofiguration of our A-DGN. We reported the average time per epoch (measured in seconds) for each model in the graph property prediction task in Table 7. Each model in the evaluation has 20 layers and an embedding dimesion equal to 30.

Table 6: Mean test set $log_{10}(\text{MSE})$ and std averaged over 4 random weight initializations for each configuration. The subscript *ws* stands for weight sharing, while *ldw* for layer dependent weights. The lower the better.

|  | Diameter | SSSP | Eccentricity |
|---|---|---|---|
| GCN | 0.7424±0.0466 | 0.9499±9.18·$10^{-5}$ | 0.8468±0.0028 |
| GAT | 0.8221±0.0752 | 0.6951±0.1499 | 0.7909±0.0222 |
| GraphSAGE | 0.8645±0.0401 | 0.2863±0.1843 | 0.7863±0.0207 |
| GIN | 0.6131±0.0990 | -0.5408±0.4193 | 0.9504±0.0007 |
| GCNII | 0.5287±0.0570 | -1.1329±0.0135 | 0.7640±0.0355 |
| DGC | 0.6028±0.0050 | -0.1483±0.0231 | 0.8261±0.0032 |
| GRAND | 0.6715±0.0490 | -0.0942±0.3897 | 0.6602±0.1393 |
| Our$_{ws}$ | -0.5188±0.1812 | -3.2417±0.0751 | 0.4296±0.1003 |
| Our$_{ldw}$ | **-0.5455±0.0328** | **-3.4020±0.1372** | **0.3046±0.1181** |
| Our$_{ws}$(GCN) | 0.2646±0.0402 | -1.3659±0.0702 | 0.7177±0.0345 |
| Our$_{ldw}$(GCN) | 0.2271±0.0804 | -1.8288±0.0607 | 0.7235±0.0211 |

Table 7: Average time per epoch (measured in seconds) and std, averaged over 4 random weight initializations. Each time is obtained by employing 20 layers and an embedding dimension equal to 30. The subscript *ws* stands for weight sharing, *ldw* for layer dependent weights.

|  | Diameter | SSSP | Eccentricity |
|---|---|---|---|
| GCN | 32.45±2.54 | 17.44±3.85 | 11.78±2.43 |
| GAT | 20.20±5.18 | 26.41±8.34 | 17.28±1.92 |
| GraphSAGE | 13.12±2.99 | 13.12±2.99 | 8.20±0.75 |
| GIN | 6.63±0.28 | 21.16±2.33 | 14.22±3.17 |
| GCNII | 13.13±6.85 | 14.96±7.17 | 15.70±3.92 |
| DGC | 8.97±9.07 | 12.54±1.62 | 7.21±11.10 |
| GRAND | 133.84±42.57 | 109.15±27.49 | 202.46±85.01 |
| Our$_{ws}$ | 8.42±2.71 | 7.86±2.11 | 13.18±9.07 |
| Our$_{ldw}$ | 14.59±8.67 | 10.47±6.95 | 14.04±11.60 |
| Our$_{ws}$(GCN) | 13.08±5.49 | 28.74±10.92 | 16.26±4.58 |
| Our$_{ldw}$(GCN) | 40.50±16.45 | 26.72±17.98 | 24.43±19.10 |

# H ADDITIONAL EXPERIMENTS

## H.1 GRAPH HETEROPHILIC BENCHMARKS

In the graph heterophilic benchmark we consider six well-known graph datasets for node classification, i.e., Chameleon, Squirrel, Actor, Cornell, Texas, and Wisconsin. We employed the same experimental setting as Pei et al. (2020). As in the graph benchmark in Section 4.2, we study only the version of A-DGN with weight sharing. We perform hyper-parameter tuning via grid search, optimizing the accuracy score. We report in Table 8 the grid of hyper-parameters explored for this experiment.

We present the results on the graph heterophilic benchmarks in Table 9, reporting the achieved accuracy. We observe that our method obtains comparable results to state-of-the-art methods on four out of six datasets (i.e., Actor, Cornell, Texas, Wisconsin). As stated in the work of Yan et al. (2021), the main cause of performance degradation in heterophilic benchmarks is strongly related to over-smoothing. Therefore, since A-DGN is not designed to tackle the over-smoothing problem, the achieved level of accuracy on these datasets is a remarkable performance. In fact, our method outperforms most of the DGNs specifically designed to mitigate this phenomenon, ranking

Table 8: The grid of hyper-parameters employed during model selection in the graph heterophilic benchmarks.

| | Configs |
|---|---|
| optimizer | Adam |
| learning rate | $10^{-1}, 10^{-2}, 10^{-4}$ |
| weight decay | $10^{-2}$ |
| n. layers | 8, 16, 32, 64 |
| embedding dim | 128, 256, 512, 1024 |
| $\sigma$ | tanh, relu |
| $\epsilon$ | $10^{-1}, 10^{-2}$ |
| $\gamma$ | $10^{-1}, 10^{-2}$ |
| dropout | 0, 0.2, 0.4, 0.6 |

third and fourth among all the models when considering the average rank of each model across all benchmarks.

Similarly to the graph benchmarks in Section 4.2, our approach maintains or improves the performance as the number of layers increases, as Figure 3 shows. Moreover, in this experiment, we show that A-DGN has outstanding performance even with 64 layers. Thus, A-DGN is able to effectively propagate the long-range information between nodes even in the scenario of graphs with high heterophilic levels. Such result suggests that the presented approach can be a starting point to mitigate the over-smoothing problem as well.

Table 9: Mean test set accuracy and std in percent averaged over different train/validation/test splits. The higher the better. The "∗" results are obtained from Yan et al. (2021), while the "⋄" results are obtained from Topping et al. (2022). We also report the average ranking of each model across all benchmarks, showing that the proposed method ranks third and fourth (in the GCN variant) among all the models considered.

| | Chameleon | Squirrel | Actor | Cornell | Texas | Wisconsin | avg rank |
|---|---|---|---|---|---|---|---|
| GGCN∗ | **71.14±1.84** | **55.17±1.58** | **37.54±1.56** | **85.68±6.63** | **84.86±4.55** | 86.86±3.29 | 1.33 |
| GPRGNN∗ | 46.58±1.71 | 31.61±1.24 | 34.63±1.22 | 80.27±8.11 | 78.38±4.36 | 82.94±4.21 | 8.50 |
| H2GCN∗ | 60.11±2.15 | 36.48±1.86 | 35.70±1.00 | 82.70±5.28 | 84.86±7.23 | **87.65±4.98** | 4.67 |
| GCNII∗ | 63.86±3.04 | 38.47±1.58 | 37.44±1.30 | 77.86±3.79 | 77.57±3.83 | 80.39±3.40 | 5.83 |
| Geom-GCN∗ | 60.00±2.81 | 38.15±0.92 | 31.59±1.15 | 60.54±3.67 | 66.76±2.72 | 64.51±3.66 | 9.17 |
| PairNorm∗ | 62.74±2.82 | 50.44±2.04 | 27.40±1.24 | 58.92±3.15 | 60.27±4.34 | 48.43±6.14 | 11.00 |
| GraphSAGE∗ | 58.73±1.68 | 41.61±0.74 | 34.23±0.99 | 75.95±5.01 | 82.43±6.14 | 81.18±5.56 | 6.67 |
| GCN∗ | 64.82±2.24 | 53.43±2.01 | 27.32±1.10 | 60.54±5.30 | 55.14±5.16 | 51.76±3.06 | 9.83 |
| GAT∗ | 60.26±2.50 | 40.72±1.55 | 27.44±0.89 | 61.89±5.05 | 52.16±6.63 | 49.41±4.09 | 11.00 |
| MLP∗ | 46.21±2.99 | 28.77±1.56 | 36.53±0.70 | 81.89±6.40 | 80.81±4.75 | 85.29±3.31 | 7.67 |
| FA⋄ | 42.33±0.17 | 40.74±0.13 | 28.68±0.16 | 58.29±0.49 | 64.82±0.29 | 55.48±0.62 | 11.33 |
| DIGL⋄ | 42.02±0.13 | 33.22±0.14 | 24.77±0.32 | 58.26±0.50 | 62.03±0.43 | 49.53±0.27 | 15.33 |
| DIGL + Undirected⋄ | 42.68±0.12 | 32.48±0.23 | 25.45±0.30 | 59.54±0.64 | 63.54±0.38 | 52.23±0.54 | 14.00 |
| SDRF⋄ | 42.73±0.15 | 37.05±0.17 | 28.42±0.75 | 54.60±0.39 | 64.46±0.38 | 55.51±0.27 | 12.67 |
| SDRF + Undirected⋄ | 44.46±0.17 | 37.67±0.23 | 28.35±0.06 | 57.54±0.34 | 70.35±0.60 | 61.55±0.86 | 11.67 |
| Our | 49.69±2.59 | 38.70±1.26 | 35.34±1.01 | 78.38±2.70 | 82.97±2.72 | 86.67±3.70 | 5.83 |
| Our(GCN) | 48.71±3.07 | 36.36±1.08 | 36.11±0.83 | 76.49±4.99 | 83.24±6.02 | 87.25±3.64 | 6.50 |

## H.2 TREE-NEIGHBORSMATCH

In the Tree-NeighborsMatch task we consider the same experimental setting as Alon & Yahav (2021). For simplicity, we focus on the version of A-DGN with weight sharing and simple ag-

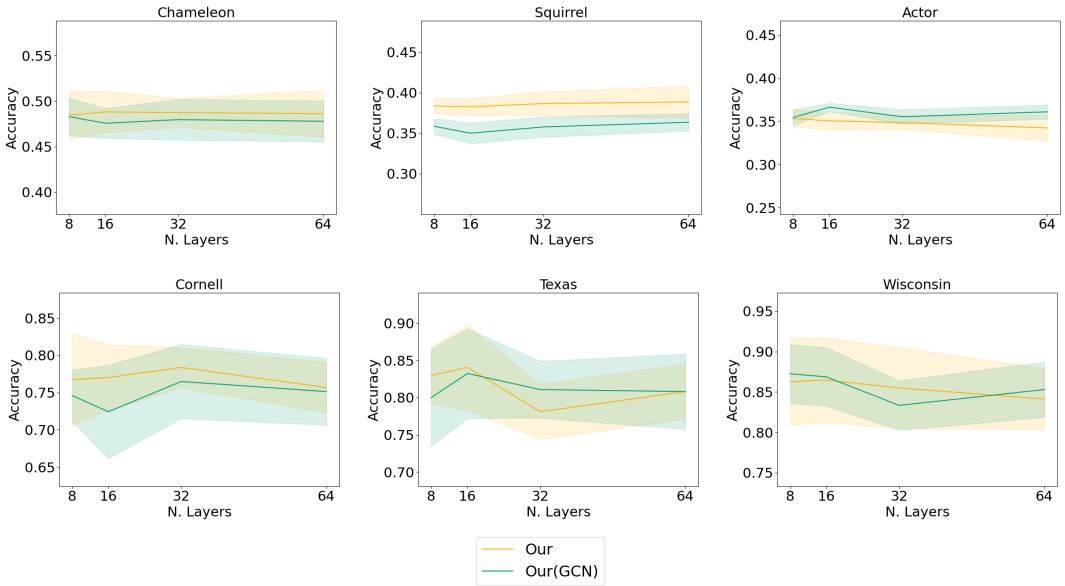

Figure 3: The test accuracy of A-DGN with respect to the number of layers on all the graph heterophilic datasets. From the top left to the bottom, we show: Chameleon, Squirrel, Actor, Cornell, Texas, and Wisconsin. The accuracy is averaged over 10 train/validation/test splits.

gregation scheme (Equation 6). We report in Table 10 the grid of hyper-parameters explored for this experiment.

Table 10: The grid of hyper-parameters employed during model selection in the Tree-NeighborMatch task.

|  | **Configs** |
| --- | --- |
| embedding dim | 32, 86, 128 |
| $\sigma$ | tanh, relu |
| $\epsilon$ | 0.1 |
| $\gamma$ | 0.1 |

We present the results on the Tree-NeighborsMatch benchmark in Table 11. In this case, to avoid having an overly unbalanced setup in terms of the number of trainable parameters, we have increased the number of hidden unit in our method (we use weight sharing, while baselines present up to 8 trainable layers). Moreover, we have normalized the achieved accuracy, reporting the ratio between the accuracy score in percentage points and the employed total number of trainable hidden units, i.e., $acc \times 100/N_{tot}$. As it can be seen, A-DGN generally outperforms the baselines (with the only exception of radius = 5). Moreover, even considering the un-normalized (i.e., original) accuracy, we observe that, even with fewer trainable parameters, A-DGN achieves on par or better performance compared to GAT, GCN, and GIN.

Table 11: Normalized train accuracy across problem radius (tree depth) in the Tree-NeighborsMatch task. The subscript in each cell contains the un-normalized (i.e., original) train accuracy. The higher the better. "N/A" denotes non-reported results. The "∗" results are obtained from Alon & Yahav (2021).

| | **Problem radius** | | | | | | |
|---|---|---|---|---|---|---|---|
| | **2** | **3** | **4** | **5** | **6** | **7** | **8** |
| GGNN∗ | $1.56_{1.0}$ | $1.04_{1.0}$ | $0.78_{1.0}$ | $0.38_{0.60}$ | $0.20_{0.38}$ | $0.09_{0.21}$ | $0.06_{0.16}$ |
| GAT∗ | $1.56_{1.0}$ | $1.04_{1.0}$ | $0.78_{1.0}$ | $0.26_{0.41}$ | $0.10_{0.20}$ | $0.07_{0.15}$ | $0.04_{0.10}$ |
| GCN∗ | $1.56_{1.0}$ | $1.04_{1.0}$ | $0.55_{0.70}$ | $0.12_{0.19}$ | $0.07_{0.14}$ | $0.04_{0.09}$ | $0.03_{0.08}$ |
| GIN∗ | $1.56_{1.0}$ | $1.04_{1.0}$ | $0.60_{0.77}$ | $0.18_{0.29}$ | $0.10_{0.20}$ | N/A | N/A |
| Our | $3.13_{1.0}$ | $3.13_{1.0}$ | $1.16_{1.0}$ | $0.32_{0.41}$ | $0.27_{0.23}$ | $0.13_{0.16}$ | $0.12_{0.10}$ |

