# OpenReview forum: "Anti-Symmetric DGN: a stable architecture for Deep Graph Networks"
_ICLR.cc/2023/Conference — ICLR 2023 poster_

### Official Review · Reviewer_Cm1J · 2022-10-22

**Confidence:** 4
**Correctness:** 2
**Technical Novelty And Significance:** 2
**Empirical Novelty And Significance:** 1
**Recommendation:** 5

**Clarity, Quality, Novelty And Reproducibility:**

\
**Clarity**

The paper is generally well-organised.

There are a few typos some of which are listed below.

* Abstract: yields to -> leads to
* First line in Section 2: networks architectures -> network architectures
* Inconsistency in the limits of the index i (d in the discussion leading upto Proposition 1 and n in Proposition 1)
* Last line in Section 2: despite we designed -> despite designing or although we designed

\
**Quality**

Experiments on oversquashing are needed to support the claim that A-DGN can alleviate/mitigate oversquashing, e.g., TreeNeighboursMatch problem in prior work [2] (Section 4.1 in the paper).

[2] On the Bottleneck of Graph Neural Networks and its Practical Implications, In ICLR'21.

For real-world datasets, the authors could consider datasets from relevant graph rewiring based methods to mitigate oversquashing [3] which are relevant baselines for A-DGN.

[3] Understanding over-squashing and bottlenecks on graphs via curvature, In ICLR'22.

More experiments are needed to confirm that A-DGN can capture long-range dependencies, e.g., the authors could consider Code2 dataset used in prior work [4] (Section 5.4 in the paper).

[4] Representing Long-Range Context for Graph Neural Networks with Global Attention, In NeurIPS'21.

\
**Novelty**

The novelty can be strengthened by positioning the contributions with prior work on oversmoothing since oversquashing, vanishing/exploding gradients and oversmoothing are associated with DGNs.

Specifically, adding/subtracting identity matrices (as in Equation 5) has been previously employed [5] for DGNs with many hidden layers in the context of oversmoothing.

[5] Simple and Deep Graph Convolutional Networks, In ICML'20.

Some of the datasets used in GRAND [6] (which has been cited) and in this paper are the same (e.g., PubMed, Coauthor CS, Photo) but GRAND has explored oversmoothing as well.

[6] GRAND: Graph Neural Diffusion, In ICML'21.

\
**Reproducibility**

The code is submitted; additionally the main part and the supplementary part include enough material, e.g., dataset details, hyperparameters, for an expert to replicate the results of the paper.
___

**Strength And Weaknesses:**

\
**Strengths**

\+ The paper is well-organised and Figure 1 clearly explains the key ideas of the proposed method.

\+ The use of anti-symmetric weights for node's own features (Equations 4 and 5) to effectively deepen DGNs is theoretically grounded and interesting.


\
**Weaknesses**

\- There are no experiments to support the claim that A-DGN can specifically alleviate/mitigate oversquashing.

\- There are no experiments to support the claim that A-DGN can effectively handle long-range dependencies specifically on graph data requiring long-range reasoning.

\- The poor long-range modelling ability of DGNs is attributed to oversquashing and vanishing/exploding gradients but the poor performance could also be due to oversmoothing, another phenomenon observed in the context of very deep graph networks [Deeper Insights into Graph Convolutional Networks for Semi-Supervised Learning, In AAAI'18].
___

**Summary Of The Paper:**

Deep graph networks (DGNs), especially those following the message passing paradigm, cannot effectively capture long-range dependencies in graph datasets and usually suffer from exploding and vanishing gradients.

This paper takes an ordinary differential equations (ODE)-perspective and proves that using anti-symmetric parameter matrices can be effective in mitigating the aformentioned issues.

The effectiveness of the resulting model, termed Anti-symmetric-DGN (A-DGN), is demonstrated on benchmark graph property prediction and node classification datasets.
___

**Summary Of The Review:**

While the proposed method is interesting, more empirical evaluation in support of the claims and positioning with relevant prior work are needed to strengthen the contributions.
___

---

> ### Author Response · Authors · 2022-11-15
> **Reply to Reviewer Cm1J**
>
> Firstly, we thank the reviewer for the comments.
>
> ***"There are no experiments to support the claim that A-DGN can specifically alleviate/mitigate oversquashing. There are no experiments to support the claim that A-DGN can effectively handle long-range dependencies specifically on graph data requiring long-range reasoning."***
>
> The three tasks proposed in Section 4.1 have the purpose of supporting the claim that our method mitigates over-squashing and propagates long-range dependencies. Indeed, all the tasks on graph property prediction require exploiting long-range dependencies in order to be solved. Clearly, to compute the diameter, the eccentricity, or the single source shortest path (SSSP) in a graph we need to propagate information along the entire graph, similarly to how standard algorithms work (e.g., Belman-Ford, Dijkstra’s algorithm). For example, in the SSSP task, we are computing the shortest paths between a given node $v$ and all other nodes in the graph. Thus, it is fundamental to propagate not only the information of the direct neighborhood of $v$, but also the information of nodes placed at the edge of the graph (which are extremely far from $v$). Similarly, for diameter and eccentricity. We have made this point more clear in the manuscript.
>
> ***"The poor long-range modelling ability of DGNs is attributed to oversquashing and vanishing/exploding gradients but the poor performance could also be due to oversmoothing, another phenomenon observed in the context of very deep graph networks [Deeper Insights into Graph Convolutional Networks for Semi-Supervised Learning, In AAAI'18]."***
>
> We agree with the reviewer that poor performances could be due also to over-smoothing, however, the employed graph benchmarks (Section 4.2) are known to have high homophily, thus we believe that over-smoothing is highly mitigated. Moreover, for each model, we select the best number of layers, which again allows traditional models (e.g., GCN, GAT) to have competitive performance. Besides that, we have included in the manuscript (Tables 1 and 2) more baselines that address the problem of over-smoothing, i.e., GRAND and GCNII.
>
> ***“There are a few typos some of which are listed below.[...]”***
>
> We thank the reviewer for having noticed the typos. We have fixed them accordingly.
>
> ***“Experiments on oversquashing are needed to support the claim that A-DGN can alleviate/mitigate oversquashing, e.g., TreeNeighboursMatch problem in prior work [2] (Section 4.1 in the paper).For real-world datasets, the authors could consider datasets from relevant graph rewiring based methods to mitigate oversquashing [3] which are relevant baselines for A-DGN.More experiments are needed to confirm that A-DGN can capture long-range dependencies, e.g., the authors could consider Code2 dataset used in prior work [4] (Section 5.4 in the paper).”***
>
> We are running the experiments on TreeNeighboursMatch, heterophile benchmark, and Code2. However, we observe that TreeNeighboursMatch and Code2 require long running times because of the large scale of the datasets. We will do our best to include such results.
>
> ***“The novelty can be strengthened by positioning the contributions with prior work on oversmoothing since oversquashing, vanishing/exploding gradients and oversmoothing are associated with DGNs.”***
>
> We thank the reviewer for the suggestion. We have included more literature to better positioning our contribution in Section 3.
>
> ***“Some of the datasets used in GRAND [6] (which has been cited) and in this paper are the same (e.g., PubMed, Coauthor CS, Photo) but GRAND has explored oversmoothing as well.”***
>
> We plan to accurately explore over-smoothing as a future research direction. However, results on the heterophily benchmark will provide us a hint if A-DGN can mitigate over-smoothing as well.

---

> > ### Author Response · Authors · 2022-11-17
> > **Manuscript update**
> >
> > ***“For real-world datasets, the authors could consider datasets from relevant graph rewiring based methods to mitigate oversquashing  [3]”***
> >
> > As suggested by the reviewer, we extended our experiments with heterophilic datasets. We report these results in Appendix G.1. In such a scenario, we performed experiments with a number of layers up to 64, confirming that A-DGN maintains or improves the performance as the number of layers increases.

---

> > > ### Author Response · Authors · 2022-11-17
> > > **Second manuscript update**
> > >
> > > ***“Experiments on [...] TreeNeighboursMatch problem and [...] Code2 dataset”***
> > >
> > > We have done our best to add the requested additional experiments, within the limited time of the rebuttal. The reviewer can now find additional experiments on the TreeNeighboursMatch problem (reported in Appendix G.2). As for the Code2 dataset timing does not allow to perform a robust and fair model selection to assess A-DGN.\
> > > Moreover, we believe that Code2 does not significantly show how A-DGN can capture long-range dependencies. Indeed, the task requires predicting the sub-tokens forming the method name, given the Abstract Syntax Tree of the source code. In such a scenario, the core information required to solve the task should be close to the root, while leaves contain only marginal information.\
> > > We believe that the experiments proposed in the manuscript are supporting the claim that our method mitigates over-squashing and propagates long-range dependencies. Indeed, as previously discussed, all the tasks on graph property prediction require exploiting long-range dependencies in order to be solved. Moreover, the experiments in Section 4.2 and Appendix G.1 confirm that A-DGN maintains or improves the performance as the number of layers increases.

---

> > > > ### Author Response · Authors · 2022-11-23
> > > > **Comment to Reviewer Cm1J**
> > > >
> > > > We thank the reviewer for their valuable feedbacks. We have taken very seriously the comments received in the first instance and dedicated the rebuttal time to run as many additional experiments requested as possible. However, we have not received any other feedback from the reviewer. We would like to know which aspects are still preventing the reviewer to consider our work worth of an accept.

---

> > > > > ### Comment · Reviewer_Cm1J · 2022-12-09
> > > > > **Thanks for the Response**
> > > > >
> > > > > Thanks to the authors for the response and the new experiments. I apologise for the delay in response.
> > > > >
> > > > > >The employed graph benchmarks (Section 4.2) are known to have high homophily, thus we believe that over-smoothing is highly mitigated
> > > > >
> > > > > Could the authors please provide any reference in support of the statement? DGNs on graph benchmarks of Section 4.2 are known to suffer from oversmoothing based on explorations in prior work [e.g., [Measuring and Relieving the Over-smoothing Problem for Graph Neural Networks from the Topological View, In AAAI'20](https://ojs.aaai.org//index.php/AAAI/article/view/5747)].
> > > > >
> > > > > > The results on the heterophily benchmark will provide us a hint if A-DGN can mitigate over-smoothing as well.
> > > > >
> > > > > The challenges on heterophily benchmarks are more than one: heterophilic edges, oversmoothing, oversquashing, etc. It is unclear how to isolate the effects of one phenomenon (e.g., oversquashing) on heterophilic data.
> > > > >
> > > > > Generally, while the proposed method is interesting and effective on benchmarks (interesting on TreeNeighboursMatch too), it is unclear where the benefits come from on real-world data (better modelling of long-range dependencies or reduced effects of other phenomena such as oversmoothing, heterophily).

---

> > > > > > ### Author Response · Authors · 2022-12-09
> > > > > > **Comment to Reviewer Cm1J**
> > > > > >
> > > > > > Dear reviewer, thank you for your feedback.
> > > > > >
> > > > > > >Could the authors please provide any reference in support of the statement? DGNs on graph benchmarks of Section 4.2 are known to suffer from oversmoothing based on explorations in prior work
> > > > > >
> > > > > > In the work by Yan et al [1], the authors show that oversmoothing is highly correlated to graph heterophily level. So it should be more difficult (even if it is not impossible) for oversmoothing to occur in high homophily conditions.
> > > > > >
> > > > > > [1] Two Sides of the Same coin: heterophily and oversmoothing in graph convolutional neural networks, Yan et al 2022
> > > > > >
> > > > > > >The challenges on heterophily benchmarks are more than one: heterophilic edges, oversmoothing, oversquashing, etc. It is unclear how to isolate the effects of one phenomenon (e.g., oversquashing) on heterophilic data.
> > > > > >
> > > > > > >Generally, while the proposed method is interesting and effective on benchmarks (interesting on TreeNeighboursMatch too), it is unclear where the benefits come from on real-world data (better modelling of long-range dependencies or reduced effects of other phenomena such as oversmoothing, heterophily).
> > > > > >
> > > > > > We have introduced experiments on heterophily benchmarks upon request of the Reviewer, and we were under the impression that these were needed by the Reviewer to gain a better understanding of the capibility of the approach. Now it is a bit puzzling to read that experiments on heterophily benchmarks do not serve the purpose of showing the benefits of the approach.
> > > > > >
> > > > > > This said, we agree with the reviewer that oversmoothing is not the only challenge in heterophilic tasks. However, as shown in the work by Yan et al., oversmoothing and heterephily levels are highly correlated. Moreover, heterophilic datasets are commonly employed to show the effectiveness against oversmoothing [1,2,3,4].  Please note that we have provided evidence of our method being effective in propagating long range dependencies in the graph property prediction tasks (Section 4.1). Indeed, in such tasks, it is fundamental to propagate information along the entire graph. Thus, we believe that it is reasonable to assume that effective information propagation is also happening when applying our approach to the other datasets (including the heterophilic ones) and that such a property is at the core of the competitive performance of our method.
> > > > > >
> > > > > > [2] PDE-GCN: Novel Architectures for Graph Neural Networks Motivated by Partial Differential Equations, Eliasof et al.  2021
> > > > > >
> > > > > > [3] Graph-Coupled Oscillator Networks, Rush et al. 2022
> > > > > >
> > > > > > [4] Neural sheaf diffusion: a topological perspective on heterophily and oversmoothing in GNNs, Bodnar et al. 2022

---

> > > > > > > ### Comment · Reviewer_Cm1J · 2022-12-09
> > > > > > > **Thanks for the Comments**
> > > > > > >
> > > > > > > Thanks for the reference on oversmoothing and heterophily [1]. One of the main contributions of that paper is that "low-degree nodes tend to trigger the oversmoothing problem in strongly homophilous graphs." Figure 2 in this submission (and Table 2 in that paper) show that models such as GCN with larger depths (e.g., 8 onwards)  degrade significantly in strongly homophilous data such as Cora, Citeseer, Pubmed.
> > > > > > >
> > > > > > > While experiments on heterophilic data are essential, the current set of real-world experiments does not give a comprehensive enough picture of capturing long-range information in light of existing work. In the camera ready / a future version, please add relevant baselines, viz., fully adjacent (FA) baseline [2] and SDRF [3] baseline in the experiments (e.g., see Table 2 in [3]). Please also add the explanantion (with compelling empirical evidence) why experiments on the Code2 dataset [4] do not significantly show how DGNs can capture long-range dependencies.
> > > > > > >
> > > > > > > The graph property prediction tasks in the submission are graph theory tasks and randomly generated datasets. It would be more compelling to see experiments on real-world data. Prior work [e.g., [4, 5]) has investigated real-world graph property prediction data in the context of modelling long-range interactions.
> > > > > > >
> > > > > > > [1] [Two Sides of the Same Coin: Heterophily and Oversmoothing in Graph Convolutional Neural Networks, In ICDM'22](https://arxiv.org/abs/2102.06462)
> > > > > > >
> > > > > > > [2] [On the Bottleneck of Graph Neural Networks and its Practical Implications, In ICLR'21](https://openreview.net/forum?id=i80OPhOCVH2)
> > > > > > >
> > > > > > > [3] [Understanding over-squashing and bottlenecks on graphs via curvature, In ICLR'22](https://openreview.net/forum?id=7UmjRGzp-A)
> > > > > > >
> > > > > > > [4] [Representing Long-Range Context for Graph Neural Networks with Global Attention, In NeurIPS'21](https://openreview.net/forum?id=nYz2_BbZnYk)
> > > > > > >
> > > > > > > [5] [Hierarchical Graph Neural Nets can Capture Long-Range Interactions, 2022](https://arxiv.org/abs/2107.07432)

---

> > > > > > > > ### Author Response · Authors · 2022-12-12
> > > > > > > > **Thanks for the comments**
> > > > > > > >
> > > > > > > > We thank the reviewer for the feedback. We can certainly add the two baselines from [3] in the camera ready. However, we respectfully disagree with the reviewer on the relevance of our experiments. In fact, we tested our model on 15 datasets, 4 of which we used specifically to show the effectiveness of long-range dependency propagation (3 synthetic tasks designed by us and the TreeNeighMatch task suggested by the reviewer). Referring to [3], previously published at ICLR, the authors show results on only 9 datasets (most of which are a subset of ours). We therefore believe that the experimental robustness and extent of our experimental analysis are in line with what is expected from this conference.

---

### Official Review · Reviewer_Hf3x · 2022-10-23

**Confidence:** 4
**Correctness:** 4
**Technical Novelty And Significance:** 3
**Empirical Novelty And Significance:** 2
**Recommendation:** 5

**Clarity, Quality, Novelty And Reproducibility:**

The paper is easy to follow and very clear.

The evaluation of the method is lacking both with comparison to existing methods and with respect to itself (e.g., an ablation study, more datasets and settings)

The work is an extension of existing method and not completely novel.

The method is sufficiently described to be reproduced.

**Strength And Weaknesses:**

Positive points:
- The work continues the research in the field of neural networks as ODEs that enables to interpret neural networks as dynamical systems.
- The authors provide nice theoretical understanding of their method.
- The paper is well organized.

Negative points:

- The concept of non-dissipative systems in GNNs is not novel and was previously proposed in the following papers:

"PDE-GCN: Novel Architectures for Graph Neural Networks Motivated by Partial Differential Equations"

"Graph-Coupled Oscillator Networks"

- It is not clear to me why the authors chose to use a layer shared W matrix and not to learn it per layer, given their claim that the theoretical analysis is also correct with such a formulation. Does it yield worse results? In any case it needs to be examined and reported.

-What is the difference between the proposed aggregation in equation (6) and the one in GraphConv (Morris et al, 2019). It seems to be the same to me.

-It is not explained in section 4.1 why did the authors choose to change the experimental settings from the original ones (number of nodes) in Corso et al.

- In the context of alleviating oversmoothing , a discussion of existing method is lacking. For example, a comparison with GCNII (Chen et al), EGNN (Zhou et al), pathGCN (Eliasof et al).

-The experimental evaluation is lacking proper comparison with recent GNN methods, for example with GraphCON (Rusch et al), GCNII (Chen et al), GRAND (Chamberlain et al).

- The authors run their experiments for up to10,000 epochs, which is a large number of epochs compared to typical methods in GNNs. Can the authors explain why? What is the effective number of epochs required to get the best results? What is the accuracy when using a smaller number of maximal epochs, like 500 or 1000 epochs?

- The scope of experiments is quite narrow. Ideally the authors should report their results on more benchmarks such as Cora/Citeseer/Pubmed and heterophilic datasets as proposed in Geom-GCN (Pei et al).

- In the context of oversmoothing evaluation the authors report the accuracy with only up to 30 layers. I think that showing the accuracy (and also possibly the energy of the node features) with more layers (e.g. 64 layers) will be more convincing.

**Summary Of The Paper:**

The authors propose an anti-symmetric GNN to tackle the oversmoothing phenomenon in GNNs.
The authors suggest an ODE based approach and analyze their model A-DGN and show that it is non-dissipative, i.e., feature/energy preserving and therefore suggest that it can alleviate the oversmoothing phenomenon. Several experiments with a variable number of layers is offered showing slight improvement over the considered methods.

**Summary Of The Review:**

The paper proposed an interesting extension of existing ODE based GNNs but lacks a discussion of them, and the experimental evaluation is lacking.

---

> ### Author Response · Authors · 2022-11-15
> **Reply to Reviewer Hf3x (part 1)**
>
> We thank the reviewer for their comments. However, we would like to observe that our method does not have the objective of tackling the over-smoothing problem. In fact, we never refer to such issue in our manuscript. We propose A-DGN to tackle the over-squashing phenomenon, which is the lack of capacity to capture information concerning interactions between nodes that are far away in the graph. Differently, over-smoothing is a phenomenon where all node features become almost indistinguishable.
> Moreover, we highlight that the obtained results show a decisive improvement with respect to baselines. Specifically, in the Graph Property prediction tasks (Section 4.1) our method improves the performance of baselines by 200% to a maximum of 300%.
>
> ***“The concept of non-dissipative systems in GNNs is not novel and was previously proposed in [1] and [2].\
> [1] PDE-GCN: Novel Architectures for Graph Neural Networks Motivated by Partial Differential Equations\
> [2]Graph-Coupled Oscillator Networks”***
>
> We thank the reviewer for the relevant related work suggested, we have integrated it into Section 3. However, even in this case, the suggested literature refers to a different problem, i.e., the over-smoothing phenomena. We agree with the reviewer that such works are designed methods with a conservative behavior, however, they are preserving the energy of the system while we are preventing that a perturbation of the input rapidly vanish during its propagation. Thus, we are preventing that long-range dependencies among nodes are forgotten. Moreover, both methods presented in [1] and [2] obtain a conservative behavior by leveraging hyperbolic equations and second-order ODEs, while our A-DGN is defined using a simpler first-order ODE and its forward Euler discretization. Therefore, we believe that the novelty of our framework is preserved since we are proposing a simple method that by design is stable, preserves long-range dependencies among nodes, and can be applied to many DGNs. Still, we thank the reviewer for the feedback, and we plan to analyze if our approach is suitable for the over-smoothing phenomena in the future.
>
> ***“It is not clear to me why the authors chose to use a layer shared W matrix and not to learn it per layer, given their claim that the theoretical analysis is also correct with such a formulation. Does it yield worse results? In any case it needs to be examined and reported.”***
>
> We report in Tables 1 and 6 the results obtained for the version of A-DGN with layer-dependent weights. As it is clear, such a configuration does not lead to worse results. We decided to use and report only the version of A-DGN with weight sharing since it achieves good performances with lower training costs.
>
> ***“What is the difference between the proposed aggregation in equation (6) and the one in GraphConv (Morris et al, 2019). It seems to be the same to me.”***
>
> We thank the reviewer for the comment. They are indeed similar, and for such a reason we included the reference to GraphConv in the manuscript).
>
> ***“It is not explained in section 4.1 why did the authors choose to change the experimental settings from the original ones (number of nodes) in Corso et al.”***
>
> We choose to change the experimental setting to make the task more challenging and increase the length of the long-range dependencies required to solve the task. We have included this clarification in Section 4.1.
>
> ***“In the context of alleviating oversmoothing , a discussion of existing method is lacking. For example, a comparison with GCNII (Chen et al), EGNN (Zhou et al), pathGCN (Eliasof et al).”***
>
> As we observed before, we are not in the context of alleviating over-smoothing. Indeed, we are focusing on the problem of propagating long-range dependencies rather than avoiding that node features converge to the same value. Despite that, we have included such literature in the manuscript (Section 3).

---

> > ### Author Response · Authors · 2022-11-15
> > **Reply to Reviewer Hf3x (part 2)**
> >
> > ***“The experimental evaluation is lacking proper comparison with recent GNN methods, for example with GraphCON (Rusch et al), GCNII (Chen et al), GRAND (Chamberlain et al).”***
> >
> > As the reviewer suggested, we included more recent baselines in our experiments. We made such results available in Tables 1 and 2. Specifically, DGC and GRAND include a comparison with neuralODE-based approaches, and GCNII includes a method that is designed to overcome the over-smoothing problem. Moreover, we observe that in the paper that proposes GRAND is stated that the model is able to overcome the problems of both over-squashing and over-smoothing, thus it is a strong baseline for our method.
> >
> > ***“The authors run their experiments for up to10,000 epochs, which is a large number of epochs compared to typical methods in GNNs. Can the authors explain why? What is the effective number of epochs required to get the best results? What is the accuracy when using a smaller number of maximal epochs, like 500 or 1000 epochs?”***
> >
> > We used the same procedure as [1] for running the experiments. To reduce the computational overhead, we decreased the number of epochs from 100k to 10k, the number of splits from 100 to 5, and the number of weight initialization from 20 to 5. This setup allows us to more accurately assess the generalization performance of different models, and does not just select the model that overfits one fixed test set. Moreover, training the models for more than 1000 epochs prevents the situation in which the maximum number of epochs stops the model while it is still learning. Lastly, the required training epochs are considerably less than our limit since we use strict early stopping. Indeed, on average, A-DGN stops after 223 epochs. Thus, the accuracy does not change if we use a maximum number of epochs of 500/1000.\
> > [1] Shchur et al. Pitfalls of Graph Neural Network Evaluation. 2018
> >
> > ***“The scope of experiments is quite narrow. Ideally the authors should report their results on more benchmarks such as Cora/Citeseer/Pubmed and heterophilic datasets as proposed in Geom-GCN (Pei et al).”***
> >
> > We have already presented results for PubMed in Table 2. Since results on graphs with high homophily rates are already included in our manuscript and the timespan of the rebuttal phase is quite narrow, we do not plan to extend our evaluation to Cora and Citeseer. Despite that, we are running experiments on the heterophilic datasets proposed in Geom-GCN.
> >
> > ***“In the context of oversmoothing evaluation the authors report the accuracy with only up to 30 layers. I think that showing the accuracy (and also possibly the energy of the node features) with more layers (e.g. 64 layers) will be more convincing.”***
> >
> > As we observed before, we are not in the context of over-smoothing. Thus, showing the energy of node features would be pointless for our method. Moreover, we believe that 30 layers are already a reasonable value to prove that our method can leverage long-range dependencies since such a value has been used also in [1]. Despite that, we will do our best to include such results according to the time left for the rebuttal.
> >
> > [1] Chamberlain et al. GRAND: Graph Neural Diffusion. 2021

---

> > > ### Author Response · Authors · 2022-11-17
> > > **Manuscript update**
> > >
> > > ***“Ideally the authors should report their results on [...] heterophilic datasets as proposed in Geom-GCN (Pei et al).”***
> > >
> > > Although we are not in the context of over-smoothing, as suggested by the reviewer, we extended our experiments with heterophilic datasets. We report these results in Appendix G.1.
> > >
> > > ***“[...] with more layers (e.g. 64 layers) will be more convincing.”***
> > >
> > > Despite the several additional experiments requested and the limited time of the rebuttal, we were able to perform additional experiments with 64 layers in the scenario of heterophilic datasets. Similarly to the graph benchmarks in Section 4.2, our approach maintains or improves the performance as the number of layers increases, as Figure 3 (Appendix G.1) shows. Moreover, in this experiment, A-DGN has outstanding performance even with 64 layers. Thus, it confirms the trend shown in Section 4.

---

> > > > ### Comment · Reviewer_Hf3x · 2022-11-18
> > > > **Discussion**
> > > >
> > > > Thank you for your rebuttal.
> > > >
> > > > I read the paper again with all your additions and I am still concerned by the following:
> > > >
> > > > 1. As another reviewer raised, the baseline of GRAND does not focus on over-squashing.
> > > >
> > > > 2. Showing the results on heterophilic datasets is not necessarily related to the network oversmoothing or not.
> > > >
> > > > 3. The added results in table 9 show a large gap between recent GNNs and yours, and also do not compare to them. For instance:
> > > >
> > > > [a] How Powerful are Spectral Graph Neural Networks.
> > > >
> > > > [b] Decoupling the depth and scope of graph neural networks
> > > >
> > > > [c] Two Sides of the Same Coin: Heterophily and Oversmoothing in Graph Convolutional Neural Networks.
> > > >
> > > > [d] Beyond Low-frequency Information in Graph Convolutional Networks.
> > > >
> > > > [e] Diverse Message Passing for Attribute with Heterophily
> > > >
> > > > All those methods achieve significantly better results and I believe that comparing with them shows a more up-to-date picture.

---

> > > > > ### Author Response · Authors · 2022-11-18
> > > > > **Response to Reviewer Hf3x**
> > > > >
> > > > > ***“As another reviewer raised, the baseline of GRAND does not focus on over-squashing.”***
> > > > >
> > > > > In the original GRAND paper, the authors refer to the over-squashing problem as bottleneck, by referring to the work of Alon et al. 2021. Moreover, the authors state in the manuscript that their “approach allows a principled development of a broad new class of GNNs that are able to address the common plights of graph learning models such as depth, oversmoothing, and bottlenecks”. Therefore, GRAND remains a solid baseline as it mitigates both oversmoothing and oversquashing. \
> > > > > We also would like to remark that the addition of GRAND was explicitly requested by this reviewer in their first review: we have taken on such a suggestion and implemented it as requested. One might wonder what has changed since that request was issued if now comparison with GRAND is considered no longer an adequate one.
> > > > >
> > > > > Alon, U. and Yahav, E. On the bottleneck of graph neural networks and its practical implications. In ICLR, 2021
> > > > >
> > > > > ***“Showing the results on heterophilic datasets is not necessarily related to the network oversmoothing or not.”***
> > > > >
> > > > > We agree with the reviewer that heterophilic datasets are not only related to the oversmoothing problem. However, the work of Yan et al 2021 [c] shows the strong correlation between heterophily levels and over-smoothing. Indeed, it is the major cause of performance degradation in heterophilic graphs. \
> > > > > And once again, the addition of the heterophilic datasets was requested by this reviewer and we complied. Again, it would be helpful to understand why such a request is no longer relevant, as we have used the limited time of this rebuttal to perform the experiments requested initially by the reviewer.
> > > > >
> > > > > [c]  Yan et al 2021, Two Sides of the Same Coin: Heterophily and Oversmoothing in Graph Convolutional Neural Networks.
> > > > >
> > > > > ***“The added results in table 9 show a large gap between recent GNNs and yours, and also do not compare to them. For instance:
> > > > > [a] How Powerful are Spectral Graph Neural Networks.\
> > > > > [b] Decoupling the depth and scope of graph neural networks\
> > > > > [c] Two Sides of the Same Coin: Heterophily and Oversmoothing in Graph Convolutional Neural Networks.\
> > > > > [d] Beyond Low-frequency Information in Graph Convolutional Networks.\
> > > > > [e] Diverse Message Passing for Attribute with Heterophily”***
> > > > >
> > > > > The experimental setting employed in the majority of the proposed work does not match ours (in terms of cleanliness of task organization and robustness of the model selection scheme), thus a direct comparison between the numbers in our paper and the ones in those work is neither fair nor possible (according to machine learning good practices). Indeed, [a] and [e] employ random splits instead of the standard setting of Geom-GCN (Pei et al), [b] does not run experiments on the heterophilic datasets, and [d] employs a sub-graph of the network. Moreover, [d] shows results only on Chameleon and Squirrel. \
> > > > > To meet the reviewer’s requests, we have integrated the results from [c] into the updated version of the paper. However, we observe that the timing did not allow us to perform a robust and fair model selection as [c] did.\
> > > > > Moreover, as previously mentioned, **our method does not have the objective of tackling the over-smoothing problem**, which is the major cause of performance degradation in heterophilic graphs (as shown in [c]). Despite we do not tackle such a problem, A-DGN shows competitive performance with SOTA methods and is clearly capable of effectively propagating the long-range information between nodes even in the more challenging scenario of graphs with high heterophilic levels.

---

> > > > > > ### Comment · Reviewer_Hf3x · 2022-11-23
> > > > > > **Thank you for the revision**
> > > > > >
> > > > > > Dear authors,
> > > > > >
> > > > > > I re-read your paper, other reviews, and all of the following discussions.
> > > > > >
> > > > > > I thank you for the detailed answers.
> > > > > >
> > > > > > Regarding comparison with GRAND: Please excuse me if my answer was misunderstood. I did not mean to claim that a comparison with GRAND is not relevant. I find it relevant as an ODE based method, similar to yours. My main point was that there are concerns raised by another reviewer GRAND is not a direct comparison to over-squashing. Although it cites Alon & Yahav, it does not provide a remedy to the over-squashing problem.
> > > > > >
> > > > > > Regarding heterophilic datasets: Here again I would like to kindly note that in our further correspondences I did not state that the additional experiments on heterophilic datasets are irrelevant, and I do appreciate that the authors added those experiments. My sole point was with respect to the the authors statement in their original response : "Although we are not in the context of over-smoothing, as suggested by the reviewer, we extended our experiments with heterophilic datasets.", it is important not to confuse with over-smoothing and heterophilic dataets.
> > > > > >
> > > > > > Regarding additional papers: I thank the authors for incorporating more recent methods into their paper.
> > > > > >
> > > > > > To summarize, I am happy with the response of the authors and therefore updated my score accordingly.

---

> > > > > > > ### Author Response · Authors · 2022-11-23
> > > > > > > **Comment to Reviewer Hf3x**
> > > > > > >
> > > > > > > We thank the reviewer for re-reading our paper and reconsidering the assessment. We would like to point out that GRAND is a direct comparison for over-squashing since the authors do not only cite the work by Alon et al. but also state that their approach is “able to address the common plights of graph learning models such as [...], and bottlenecks” (i.e., over-squashing). Regarding the heterophilic datasets, it was not our intention to mix the two aspects, but we are glad that the request to add  heterophilic tasks has allowed us to show that our approach can also tackle such problems effectively.
> > > > > > > We are glad to have engaged in this interesting discussion but, having answered to all weak points raised in the review, we would like to know which aspects are still preventing the reviewer from considering our work worthy of an accept.
> > > > > > >
> > > > > > > [c]  Yan et al 2021, Two Sides of the Same Coin: Heterophily and Oversmoothing in Graph Convolutional Neural Networks.

---

> > > > > > > > ### Comment · Reviewer_Hf3x · 2022-11-29
> > > > > > > > **Discussion**
> > > > > > > >
> > > > > > > > Dear authors,
> > > > > > > >
> > > > > > > > I thank you for the thorough discussion and rebuttal. I agree with your claim about the written discussion in the introduction of GRAND and the citations of Alon & Yahav regarding over-squashing. However, I also agree with reviewer fEsH that
> > > > > > > > that GRAND does not offer a remedy to the over-squashing problem. (and yet, I think it is a valuable comparison as-is)
> > > > > > > >
> > > > > > > > Overall, I do think that the rebuttal addressed some concerns, and I therefore increased my score. I still do not think that the method merits an acceptance at this stage because of the empirical results. They are quite far from recent SOTA methods.

---

> > > > > > > > > ### Author Response · Authors · 2022-11-30
> > > > > > > > > **Discussion**
> > > > > > > > >
> > > > > > > > > Dear reviewer Hf3x,\
> > > > > > > > > We thank you for the engaged discussion. We still believe that our empirical assessment shows that our method strongly outperforms SOTA methods in what concerns exploiting long-range dependencies between nodes, which is the main objective of our method. Indeed, in the graph property prediction tasks, A-DGN is up to 300% better than literature methods. Interestingly, A-DGN achieves better or on-par performance even on those problems where the exploitation of long-range dependencies is not fundamental (as evident from Table 2 in Section 4.2 "Graph Benchmarks"). Regarding the additional experiments in Table 9, which are outside the objectives of our proposal (and despite the reduced model selection we had to implement to comply with the timing constraints), the achieved performance is still among the top 3 performing methodologies compared to the literature (including approaches that, differently from ours, were specifically designed for the peculiarities of those tasks). Based on these facts, we believe that the experimental analysis shows the flexibility and effectiveness of our proposed method, and not that A-DGN is far from SOTA baselines.

---

> > > > > ### Comment · Area_Chair_Rnx6 · 2022-11-23
> > > > > **Please stay consistent with your arguments**
> > > > >
> > > > > Dear reviewer,
> > > > >
> > > > > You gave a strongly negative score to the paper and provided some arguments. Now that the authors have responded to these comments, including a comparison to GRAND which you asked for, you suddenly agree with another reviewer that the comparison to GRAND isn't actually useful. You proceed to simply list some papers and write "the authors should compare to these". You don't provide a reason for why exactly and simply leave your score as it is. I strongly disagree with this type of review. It seemingly tries to find reasons to reject a paper, without engaging with the arguments of the authors. If you don't clarify your remaining criticism beyond a list of related papers without comment, I won't be able to take your review into account when writing the meta review.
> > > > >
> > > > > Thank you for your understanding.

---

> > > > > > ### Comment · Reviewer_Hf3x · 2022-11-23
> > > > > > **Thank you for the discussion**
> > > > > >
> > > > > > Dear Area Chair,
> > > > > >
> > > > > > Thank you for continuing the discussion. In my review (and following discussion) I tried to also be aware of other reviewers opinions and comments. I think this is a healthy reviewing process because I may have missed points that were raised by other reviewers.
> > > > > > Also as the discussion period with the authors is past, I took the liberty to take a bit more time and re-read the paper and the complete reviews and rebuttal again, therefore I did not reply to this point.
> > > > > >
> > > > > > Regarding the additional papers that I asked the authors to discuss and compare to (because those papers are recent and show higher accuracy figures than in this paper) - I think that this is a fair point and I also see that in the response from the authors, they now discuss those papers.
> > > > > >
> > > > > > I will reply to the authors and update my review accordingly.

---

### Official Review · Reviewer_fEsH · 2022-10-24

**Confidence:** 3
**Correctness:** 3
**Technical Novelty And Significance:** 3
**Empirical Novelty And Significance:** 2
**Recommendation:** 6

**Clarity, Quality, Novelty And Reproducibility:**

Clarity

I would say that the statement of Proposition 1 is not rigorous. For example, $\approx 0$ is an informal statement. Also, non-dissipative is not mathematically defined. Therefore, I would suggest writing the formal statement of Proposition 1. However, if there is a risk that it would be too complicated to include in the main text, it would be acceptable to include only informal results in the main text and a precise statement in the appendices.
Nevertheless, this paper is clear overall, and I can understand the main point of the paper.


Quality

One of the objectives of this paper is to solve the over-squashing problem (and the performance degradation of GNN caused by the problem). However, the numerical experiments only considered the GNNs proposed prior to [Alon & Yahav, 2021], which pointed out the over-squashing problem. Therefore, I would suggest comparing the proposed method with existing methods that addressed these problems (e.g., [Topping et al., 2022]). Also, the performance degradation of deep GNNs is the subject of interest for the over-smoothing problem. Therefore, I suggest comparing the proposed method with methods that mitigate the over-smoothing problem (e.g., GCNII [Chen et al., 2020]).

[Topping et al., 2022]: https://openreview.net/forum?id=7UmjRGzp-A
[Chen et al., 2020] https://proceedings.mlr.press/v119/chen20v.html


Novelty

Several existing studies have analyzed GNNs by considering an ODE obtained as its continuum limit. However, to the best of my knowledge, few studies have effectively utilized the benefit of converting GNNs to ODEs. This study related the problem of over-squashing to the properties of the corresponding ODEs and derived a desirable property of GNNs, clarifying the connection between GNNs and ODEs.

Reproducibility

OK. This paper provides the code for the numerical experiments as supplement material. Although I have not run the code, I expect that it is possible to reproduce the same experiments as the authors.

**Details Of Ethics Concerns:**

N.A.

**Strength And Weaknesses:**

Strengths
- The proposed method provides a new approach to the over-squashing problem by associating a GNN with the properties (stability, non-dissipative) of corresponding ODEs.
- The proposed method is easy to implement and can be applied to many GNNs.
- The proposed method can effectively mitigate the accuracy degradation caused by deepening GNNs (Figure 2).

Weaknesses
- The mathematical statement is somewhat informal.
- Comparison with baseline methods that addressed the over-squash problem or performance degradation of deep is preferable.

**Summary Of The Paper:**

This paper proposed an anti-symmetric deep graph neural network (A-DGN) to solve the over-squashing problem of GNNs. First, this paper gave a sufficient condition that an ODE corresponding to a given GNN is stable and non-dissipative (Proposition 1). Then, to satisfy the sufficient condition, they proposed to make the GNN weights anti-symmetric (Proposition 2), deriving A-DGN. Finally, numerical experiments about the graph property prediction problems and the graph classification problems were conducted to verify the usefulness of the proposed method.

**Summary Of The Review:**

This paper provides theoretically grounded solutions to the problem of accuracy degradation due to the deepening of GNNs from the standpoint of ODE stability. The method is simple and applicable to a wide range of GNNs. On the other hand, the numerical experiments have not been compared with recent GNN models, and comparing the proposed method with them is desirable.

---

> ### Author Response · Authors · 2022-11-15
> **Reply to Reviewer fEsH**
>
> ***“The mathematical statement is somewhat informal.”***
>
> We thank the reviewer for the comment. In Proposition 1, the idea is that if the real part of the eigenvalues of the Jacobian matrix are =0 then both forward and backward propagations are stable and non-dissipative. However, if we are extremely close to 0 we obtain the same behavior but slightly mitigated. However, in such a scenario, the level of stability and non-dissipation can only be measured experimentally. To make it more clear, we replaced ≈0 with =0. Moreover, we will report a mathematical definition of non-dissipative behavior.
>
> ***“Comparison with baseline methods that addressed the over-squash problem or performance degradation of deep is preferable.”***
>
> As the reviewer suggested, we included more baselines in our experiments. We made such results available in Tables 1 and 2. Specifically, DGC and GRAND include a comparison with neuralODE-based approaches, and GCNII includes a method that is designed to tackle the over-smoothing problem. Moreover, we observe that GRAND is designed to address over-squashing and over-smoothing problems, thus it is a strong baseline for our method. Lastly, the employed graph benchmarks (Section 4.2) are known to have high homophily, thus we believe that over-smoothing is highly mitigated. Moreover, for each model, we select the best number of layers, which again allows traditional models (e.g., GCN, GAT) to have competitive performance.

---

> > ### Author Response · Authors · 2022-11-17
> > **Manuscript update**
> >
> > ***“non-dissipative is not mathematically defined”***
> >
> > We thank the reviewer for the suggestion, we have now included a definition of non-dissipative system in Appendix A.

---

> > > ### Comment · Reviewer_fEsH · 2022-11-17
> > > **Response to authors' comments**
> > >
> > > I thank the authors for replying to my comment.
> > >
> > > **About mathematical statements**
> > >
> > > I thank the authors for the explanation. The mathematical meaning of Proposition 1 is more precise than the original version.
> > >
> > > However, I would say the proof of Proposition 1 needs modification. For example, the proof adds the assumption that Jacobian does not change significantly over time, which does not appear in the statement.
> > > In addition, the proof is still informal. Take the claim for the case $\mathrm{Re}(\lambda_i(J(t))) \ll 0$ for all $i$, although it gives an intuitive idea, it would be difficult to think of it a rigorous proof:
> > >
> > > > input perturbations would rapidly vanish, hence the nodes' representations would be insensitive to differences in the input graph, and the system would be dissipative
> > >
> > > Also, I have an additional question about the proof of Proposition 2. Specifically, the Jacobian $J(t)$ in Eq. (11) does not take the derivative of $X(t)$ in $\Phi$ into account (there should be a factor related to the derivative of $\Phi$).
> > >
> > > In view of this, I would suggest rechecking the proofs of propositions.
> > >
> > >
> > > **Comparison with baseline methods**
> > >
> > > I thank the authors for adding GCNII results. I understand the proposed method is on par with the modern GNN that addressed the over-smoothing problem. On the other hand, although GRAND is a strong baseline worth being compared, if I am not wrong, it is not a model for the over-squashing problem. In fact, the original GRNAD paper [Chamberlain et al., 2021] did not refer to the over-squashing problem. So, I want to clarify this part.

---

> > > > ### Author Response · Authors · 2022-11-18
> > > > **Response to Reviewer fEsH**
> > > >
> > > > ***“I would say the proof of Proposition 1 needs modification. [...] Also, I have an additional question about the proof of Proposition 2. J(t) in Eq. (11) does not take the derivative of X(t) in $\Phi$ into account”***
> > > >
> > > > Thank you very much for these constructive comments. Proposition 1 and its proof have been clarified in the revised paper. Specifically, following the reviewer’s suggestion, we have now included in the statement of the proposition the assumption that the Jacobian does not change significantly over time. We have also worked on making the proof more formal, as requested. In particular, the claim for the case of $Re(\lambda_i(J(t))) \ll 0$ has now been formulated more formally and more generally for the case of $Re(\lambda_i(J(t))) < 0$, making the description more precise.\
> > > > In the new version of the paper, we have revised the mathematical content also regarding Proposition 2, taking into account the derivative of x(t) in $\Phi$, and extending the analysis in this sense (revising Proposition 2 and adding some more comments in a new Appendix section D “Bounded eigenvalues of J(t)”).
> > > > In particular, we find that, when the aggregation function $\Phi$ is independent of $x_u$ (as in Equation 6), the Jacobian of the resulting ODE has purely imaginary eigenvalues, hence it is stable and non-dissipative according to Proposition 1. The statement of Proposition 2 has been modified accordingly to be more precise. Whenever $\Phi$ depends on $x_u$ (as in Equation 7), we find that the eigenvalues of J(t) are still bounded in a neighborhood around the imaginary axis as a consequence of the Bauer-Fike’s theorem, as discussed in the added Appendix D of the revised paper.
> > > >
> > > > We would like to overall thank the reviewer for the useful comments and the stimulus to better report the mathematical analysis provided in the paper, which we believe is now decisively improved in the revised version.
> > > >
> > > >
> > > > ***“I understand the proposed method is on par with the modern GNN that addressed the over-smoothing problem. ”***
> > > >
> > > > Based on our updated results on the heterophilic benchmarks, we can notice that even though A-DGN is not specifically designed for tackling the over-smoothing problem, it outperforms most of the DGNs designed for such a purpose. Moreover, we would like to observe that A-DGN is not on par with modern GNN on the other tasks. In the most relevant scenario of the graph property prediction (Section 4.1), our method shows a decisive improvement with respect to baselines. Specifically, it achieves a performance that is 200% to 300% better than the best baseline in each task.
> > > >
> > > > ***“On the other hand, although GRAND is a strong baseline worth being compared, if I am not wrong, it is not a model for the over-squashing problem. In fact, the original GRNAD paper [Chamberlain et al., 2021] did not refer to the over-squashing problem. So, I want to clarify this part.”***
> > > >
> > > > In the original GRAND paper, the authors refer to the over-squashing problem as bottleneck, by referring to the work of Alon et al. 2021. Moreover, the authors state in the manuscript that their “approach allows a principled development of a broad new class of GNNs that are able to address the common plights of graph learning models such as depth, oversmoothing, and bottlenecks”. Therefore, GRAND remains a solid baseline as it mitigates both oversmoothing and oversquashing.
> > > >
> > > > Alon, U. and Yahav, E. On the bottleneck of graph neural networks and its practical implications. In ICLR, 2021

---

> > > > > ### Comment · Reviewer_fEsH · 2022-12-02
> > > > > **Second Response**
> > > > >
> > > > > **Proposition 1**
> > > > >
> > > > > I would say there are still some informal aspects in the statement and proof of Proposition 1. For example, I wonder how to define "$J(t)$ does not change significantly over time" mathematically. However, I think I understand what the idea of this proposition.
> > > > >
> > > > > **Proposition 2**
> > > > >
> > > > > OK
> > > > >
> > > > > **Heterophilic setting**
> > > > >
> > > > > I appreciate that the authors shared the results of the Heterophilic graph, even though they are somewhat negative. Figure 2 shows that the proposed method mitigated the over-smoothing phenomena.
> > > > > Existing studies observed that over-smoothing is related to performance degradation on heterophilic graphs. Therefore, I expected that the proposed method would perform well on heterophilic graphs. I think it is worth investigating the reason for the performance degradation on the Chamelon and Squirrel datasets.
> > > > >
> > > > > **GRNAD and over-squashing**
> > > > >
> > > > > I am sorry that I did not realize that the GRAND paper [Chamberlain et al., 2021] did refer to [Alon and Yahav, 2021], which raised the over-squashing problem. However, I am not sure whether we can safely say that GRAND is a solution for the over-squashing.
> > > > > On the one hand, GRAND is similar to the Fully-Adjacent layer proposed by [Alon and Yahav, 2021] in that they performed the graph rewiring. It was experimentally confirmed (Section 6.3 of [Chamberlain et al., 2021]) that the performance of GRAND-rw is better when the rewired graph is dense. It could be related to over-squashing. On the other hand, GRAND did not state (at least explicitly) that it is a solution to the over-squashing problem.

---

### Official Review · Reviewer_29BN · 2022-10-25

**Confidence:** 4
**Correctness:** 4
**Technical Novelty And Significance:** 3
**Empirical Novelty And Significance:** 3
**Recommendation:** 8

**Clarity, Quality, Novelty And Reproducibility:**

Overall, I enjoyed reading the paper. The different concepts are clearly presented. At the same time, the paper keeps a good balance between theoretical contributions and empirical evaluation.  Despite building on existing models that first studied the relationship between GNNs and ODEs, the paper still presents several novel ideas.

Next, I highlight some of my concerns, in addition to the main limitation above, that need further clarification.

* To what extend do the datasets used contain long-range dependencies? I understand that this is the case while predicting the diameter of the graph, but this is not obvious in the real-world datasets used in the paper.

* The paper does not discuss the running time of the proposed methodology. I understand that the focus is on capturing long-range dependencies, but having an image of the time complexity would be helpful for the reader.


**Strength And Weaknesses:**

Strengths:
* The paper provides an elegant formulation of GNNs based on ODES. More importantly, several message aggregation schemes can be incorporated into the model.

*  Providing theoretical arguments about the properties of A-DGN is a key part of the paper. Even if some properties can be derived from previous studies (including works in control theory), they still contribute to the understanding of the model.

* Good experimental pipeline, considering different important tasks.

Weaknesses:
* The main limitation of the paper is the choice of baseline models. While the paper focuses on addressing the over-squashing effect, none of the baselines has specifically been designed to do so. Therefore, to this extend, the part of the evaluation does not favor traditional GNNs.

  Moreover, while training deep traditional models (e.g., GCN, GAT), the problem of over-smoothing occurs. From the discussion, this has not been taken into account in the evaluation. For instance, methodologies like PairNorm could be used to prevent over-smoothing.

  At the same time, no prior work on neural ODEs is considered. I would expect that some of these models (e.g., DGC, GRAND) should have been part of the evaluation



**Summary Of The Paper:**

The paper introduces a new formulation of GNNs based on ODEs. The main goal of the method is to ensure stability and non-dissipation properties, thus allowing the model to preserve long-range dependencies with deep architectures that also avoid vanishing gradient problems. The paper provides theoretical arguments about the properties of the proposed A-DGN model. Moreover, experiments on several tasks support the benefit of the A-DGN over several baseline models.

**Summary Of The Review:**

Overall, I believe it’s an interesting approach that further increases our understanding of the relationship between GNN models and ODEs. Some aspects related to the experimental evaluation and the choice of baseline models should be addressed.

---

> ### Author Response · Authors · 2022-11-15
> **Reply to Reviewer 29BN**
>
> First, we would like to thank you for the feedback. Below, we address each concern individually.
>
> ***“The main limitation of the paper is the choice of baseline models. While the paper focuses on addressing the over-squashing effect, none of the baselines has specifically been designed to do so. Therefore, to this extend, the part of the evaluation does not favor traditional GNNs. Moreover, while training deep traditional models (e.g., GCN, GAT), the problem of over-smoothing occurs. From the discussion, this has not been taken into account in the evaluation. For instance, methodologies like PairNorm could be used to prevent over-smoothing. At the same time, no prior work on neural ODEs is considered. I would expect that some of these models (e.g., DGC, GRAND) should have been part of the evaluation”***
>
> As the reviewer suggested, we included new baselines in our experiments. We made such results available in Tables 1 and 2. Specifically, DGC and GRAND include a comparison with neuralODE-based approaches, and GCNII includes a method that is designed to overcome the over-smoothing problem. Moreover, we observe that in the paper that proposes GRAND it is stated that the model is able to overcome the problems of both over-squashing and over-smoothing, thus it is a strong baseline for our method. Lastly, the employed graph benchmarks (Section 4.2) are known to have high homophily, thus we believe that over-smoothing is highly mitigated. Moreover, for each model, we select the best number of layers, which again allows traditional models (e.g., GCN, GAT) to have competitive performance.
>
> ***“To what extend do the datasets used contain long-range dependencies? I understand that this is the case while predicting the diameter of the graph, but this is not obvious in the real-world datasets used in the paper.”***
>
> All the tasks of graph property prediction (Section 4.1) require exploiting long-range dependencies in order to be solved. Indeed, to compute the diameter, the eccentricity, or the single source shortest path (SSSP) in a graph we need to propagate information along the entire graph, similarly to how standard algorithms work (e.g., Belman-Ford, Dijkstra’s algorithm). For example, in the SSSP task, we are computing the shortest paths between a given node $v$ and all other nodes in the graph. Thus, it is fundamental to propagate not only the information of the direct neighborhood of $v$, but also the information of nodes placed at the edge of the graph (which are extremely far from $v$). Similarly, for diameter and eccentricity. \
> We included the graph benchmarks to demonstrate that our proposal generally compares well with literature methods (outperforming them in some cases) even on datasets for which there is no evidence to support the increased propensity of the architectural bias introduced by the paper (i.e., long-range dependencies in the graphs). As a side observation, our results on these benchmarks indicate that a number of layers > 5 can lead to improved performance, hence suggesting the actual role of long-range dependencies also in such cases.
>
> ***“The paper does not discuss the running time of the proposed methodology. I understand that the focus is on capturing long-range dependencies, but having an image of the time complexity would be helpful for the reader.”***
>
> We thank the reviewer for the suggestion. We have included such information in Table 7 (Appendix E).

---

### Author Response · Authors · 2022-11-15
**General Comment**

We thank the reviewers for their feedback and comments. We have done our best to address them below, and we will constantly update the manuscript until we collect all the results. Furthermore, we also thank the reviewers for finding important positive points in our submission, including that our work further increases the understanding of the relationship between GNN models and ODEs; the proposed method mitigates the over-squashing problem by associating a GNN with stability and non-dissipative properties of the corresponding ODE (which effectively mitigates the accuracy degradation caused by deepening GNNs) and it can be applied to any GNN; lastly the explanation clarity of our paper.

We observe that all the reviewers have requested many experiments. Although we will do our best to include as many experiments as possible, the time required to run all the experiments, the timespan of the rebuttal phase, and the space left in the manuscript do not allow us to include them all.

We have already updated some of the comments and results in the manuscript (highlighted in red in this version), but we plan to constantly update the manuscript as we collect more results until the rebuttal deadline.

_We now address each of the reviews individually below._

---

### Comment · Area_Chair_Rnx6 · 2022-11-16
**Please read and respond to the rebuttals**

Dear reviewers,

The authors have responded to your reviews. Since the cannot update the submission after Nov 18, please make an effort to engage with them as soon as possible. Thank you very much!

AC

---

### Decision · Program_Chairs · 2023-01-20

**Decision:**

Accept: poster

**Justification For Why Not Higher Score:**

There is room for improvement in the way formal properties of the proposed approach were derived. Some reviewers were still not fully convinced about the experiments and requested additional comparisons to baseline methods addressing the oversquashing problem.

**Justification For Why Not Lower Score:**

Most reviewers saw merit in the work and it provides a novel perspective on deriving GNNs as discretized ODEs. Especially one reviewer was very positive about the paper.

**Metareview: Summary, Strengths And Weaknesses:**

The authors propose Anti-Symmetric Deep Graph Networks (A-DGNs) which is a framework for stable and non-dissipative deep graph networks (DGNs are often called graph neural networks). The framework is based on a formulation based on ODEs and several message aggregation schemes can be incorporated into the model. Most reviewers thought that the paper was interesting and provided a novel perspective on GNNs. The main criticism was about (a) the theoretical results in the paper: there was/is room for improvement in making the statements more formal and rigorous (b) the experimental results addressing the claimed advantage of the method to avoid over-squashing.  Both sides brought up good arguments and the authors provided additional experiments in response to criticism (b) and improved (a).



**Note From Pc:**

if the above contains the word "oral" or "spotlight" please see: "oral" presentation means -> notable-top-5% and "spotlight" means -> notable-top-25%. As stated in our emails, we are disassociating presentation type from AC recommendations